# Deleterious heteroplasmic mitochondrial mutations are associated with an increased risk of overall and cancer-specific mortality

Yun Soo Hong[1,21], Stephanie L. Battle [1,2,21], Wen Shi [1], Daniela Puiu[3], Vamsee Pillalamarri[1], Jiaqi Xie[1], Nathan Pankratz [4], Nicole J. Lake[5,6], Monkol Lek [5], Jerome I. Rotter [7], Stephen S. Rich [8], Charles Kooperberg [9], Alex P. Reiner [9], Paul L. Auer[10], Nancy Heard-Costa [11,12], Chunyu Liu[12,13], Meng Lai [13], Joanne M. Murabito[14], Daniel Levy[15], Megan L. Grove [16], Alvaro Alonso [17], Richard Gibbs[18], Shannon Dugan-Perez[18], Lukasz P. Gondek [19], Eliseo Guallar[20] & Dan E. Arking [1] ✉

Mitochondria carry their own circular genome and disruption of the mitochondrial genome is associated with various aging-related diseases. Unlike the nuclear genome, mitochondrial DNA (mtDNA) can be present at 1000 s to 10,000 s copies in somatic cells and variants may exist in a state of heteroplasmy, where only a fraction of the DNA molecules harbors a particular variant. We quantify mtDNA heteroplasmy in 194,871 participants in the UK Biobank and find that heteroplasmy is associated with a 1.5-fold increased risk of all-cause mortality. Additionally, we functionally characterize mtDNA single nucleotide variants (SNVs) using a constraint-based score, mitochondrial local constraint score sum (MSS) and find it associated with all-cause mortality, and with the prevalence and incidence of cancer and cancer-related mortality, particularly leukemia. These results indicate that mitochondria may have a functional role in certain cancers, and mitochondrial heteroplasmic SNVs may serve as a prognostic marker for cancer, especially for leukemia.

Mitochondria are the main energy production organelle of the cell and are also involved in numerous cellular processes including calcium homeostasis, apoptosis, fatty acid oxidation, and the generation of metabolic intermediates[1]. Mitochondria contain their own circular chromosome (mtDNA) that is present in 1000 s to 10,000 s copies per somatic cell[1] and is replicated independently of the nuclear genome. Its lack of protective histones, limited DNA repair system, and proximity to reactive oxygen species (ROS) producing reactions result in the acquisition of mutations in a fraction of mtDNA molecules within a cell or tissue, a state known as heteroplasmy. Heteroplasmic single nucleotide variants (SNVs) are responsible for mitochondrial disorders when present at high allele frequencies[1] and most pathogenic mtDNA SNVs are only observed in a heteroplasmic, as opposed to homoplasmic, state[2].

Mitochondrial heteroplasmy is common, with low level heteroplasmy observed in whole-genome sequencing (WGS) studies in up to 40–45% of samples[3–5]. Approximately 30% of heteroplasmies observed in any given individual are maternally inherited[6]. The effect of accumulating somatic mtDNA mutations contributes to mitochondrial dysfunction, with the most severe effects observed in tissues with the highest energy demands[7], and the role of mitochondrial dysfunction in longevity, cancer, and degenerative diseases has been well-established[3,8]. Better understanding of the effect of the presence and level of mitochondrial heteroplasmy on common or rare diseases will provide insight into the molecular physiology of disease development and progression. Large population-based cohorts, like the UK Biobank (UKB), are useful tools to tackle this problem, particularly for evaluating the association of heteroplasmy with common diseases.

---

We developed a bioinformatics pipeline, MitoHPC, to accurately measure mtDNA SNVs in large WGS datasets (https://github.com/ArkingLab/MitoHPC)[9]. MitoHPC's key feature is that it constructs a consensus mitochondrial sequence for each individual, which allows for more accurate read mapping and for measuring heteroplasmy against an individual's unique mitochondrial genome. MitoHPC performs two iterations of variant calling, first to identify major alleles, or homoplasmies, which are used to construct the consensus mitochondrial sequence, and second to call heteroplasmic variants. On simulated data, MitoHPC accurately identified all heteroplasmic SNV variants without false-positives or false-negatives. MitoHPC is also built to handle large datasets of 100,000 s of samples, making this an ideal pipeline for our study.

Here we present an analysis of mtDNA heteroplasmy measured in 194,871 individuals from the UK Biobank, a large cohort of men and women aged 40–69 years recruited in 2006–2010 with uniformly collected health information and biological materials[10]. We characterize mtDNA SNVs based on their genetic features and investigate the relationship between SNVs at highly constrained sites and risk of all-cause and cause-specific mortality. We also validate the results in additional cohorts in Trans-Omics for Precision Medicine (TOPMed) program. We find disease phenotypes that are associated with mtDNA heteroplasmic SNVs and propose the use of mitochondrial heteroplasmic SNVs detected in the blood as a risk marker for hematological cancers.

## Results

### Heteroplasmic SNV characteristics

After quality control procedures (Supplementary Fig. 1 and Supplementary Fig. 2), we identified a total of 74,369 heteroplasmic SNVs at a 5% variant allele fraction (VAF) in the UKB with 59,414 (30.5%) out of 194,871 participants having at least 1 heteroplasmy (Table 1, Fig. 1). This includes 11,602 unique variant alleles occurring at 10,161 (61.8%) out of 16,443 possible mtDNA base positions (polyC homopolymer regions are excluded due to challenges in sequencing accurately). Of the 11,602 heteroplasmic SNVs, 4257 (36.7%) were seen in only a single individual and the vast majority were extremely rare, with 10,297 (88.8%) seen in ≤10 participants (Table 2).

Alleles present in 95% or more of the mtDNA WGS reads within a participant were counted as homoplasmic SNVs. A total of 4,540,598 homoplasmic SNVs were identified, with 385 individuals having no homoplasmic variants (i.e., variants that completely match the rCRS reference mtDNA genome). Homoplasmic SNVs occurred at 7929 unique base positions, and at 6586 (83.1%) of those sites we also observed a heteroplasmic allele. Overall, there were 14,285 unique variants found at 11,318/16,569 mtDNA sites (68.3%) (Fig. 2a), with 49.3% of SNVs seen as both heteroplasmic and homoplasmic, 31.9% heteroplasmic only, and 18.8% homoplasmic only (Fig. 2b). Thus, while most heteroplasmies are extremely rare, the majority (~60%) are also found as homoplasmies.

To better understand mitochondrial mutational load, we investigated the characteristics of the heteroplasmic SNVs. We observed a transition-to-transversion (Ti/Tv) ratio of 28.7 for heteroplasmies (Table 3) which is also observed in PhyloTree sequences[11] and likely due to misincorporation by the mitochondrial polymerase gamma[12]. This is in contrast to Ti/Tv ratio in the nuclear genome, which is typically around two[13]. The overall distribution of SNVs by complex/region highlights tRNA genes as less tolerant to variation, with only 23.0% of possible SNVs observed, in contrast to Complex V, with 41.8% of possible SNVs observed (P value $5.0 \times 10^{-62}$; Fig. 2c, d). The overall distribution of the median VAFs for SNVs seen as heteroplasmies is lower than SNVs seen as both hetero- and homoplasmy (Fig. 2e). We hypothesized that SNVs with a functional change would be more likely to have low VAF in a given participant, as these SNVs would likely alter protein function and be under negative selection. Across all

participants, we see 4,252 nonsynonymous and 3,946 synonymous SNV sites, with nonsynonymous SNVs having a lower median VAF than synonymous SNVs (P value $5.4 \times 10^{-60}$; Fig. 2f and Table 3). When stratified by haplogroup (L, M, N, R, R0, U, JT, and H), the number of heteroplasmies significantly differed by haplogroups, with haplogroups L, M, and R, on average, having lower heteroplasmy count compared to haplogroup H (Bonferroni-corrected P values 0.003, 0.026, and 0.003, respectively; Fig. 2g, h). However, after adjusting for potential confounders between haplogroups, including age, sex, center, and smoking status, these groups were no longer significantly different from haplogroup H (all Bonferroni-corrected P value > 0.05). A global test for differences in heteroplasmy count by haplogroup was nominally significant (P = 0.042), with 0.007% of the variance of heteroplasmy count explained by haplogroups in the adjusted model.

Of the SNVs in Complex I genes, 53.5% were synonymous SNVs and these genes had the smallest fraction of nonsynonymous SNVs (dN/dS ratio) compared to other complexes (Bonferroni-corrected P value $8.8 \times 10^{-16}$; Table 3). Complex I facilitates the first step of oxidative phosphorylation and deficiencies or inhibition of Complex I lead to increased ROS, reduced NADP levels, and disrupt mitochondrial morphology[14,15]. Interestingly, Complex V genes had the highest dN/dS ratio at 1.92 (Bonferroni-corrected P value $6.2 \times 10^{-5}$). Complex V, also known as the ATP synthase, is the ATP generating component of the electron transport chain (ETC) and we would expect SNVs in these genes to impact ATP production.

We observed 49 different nonsense heteroplasmic mitochondrial SNVs (encoding stop codons) (Table 3) across 85 individuals, with no individual harboring more than one nonsense SNV. One of these variants, m.3308 T > G, was also found as a homoplasmic variant at a frequency of ~1:1000, consistent with previous reports of this variant[16]. This variant is likely more tolerated than other nonsense mutations, as variant m.3308 T > G is at the start of *MT-ND1*, and the methionine at codon 3 is a likely alternative translation start site, resulting in a loss of only the first 2 amino acids. One additional homoplasmic nonsense variant was found only as a singleton, m.15327 C > G near the end of *MT-CYB*. This variant was not reported in the gnomAD or Helix database, but is reported in 1 individual in Mitomap, thus demonstrating the strong selection against homoplasmic nonsense SNVs. The majority of the heteroplasmic nonsense mutations occurred at 28 sites in Complex IV genes (*MT-CO1*, *MT-CO2* and *MT-CO3*), with a marked lower frequency in Complex I genes (P value $1.6 \times 10^{-5}$; Table 3). Only 14 nonsense SNVs were seen more than once in the dataset, with the highest frequency SNV seen in 16 individuals. Only 1 individual had VAF > 31%, with a median VAF of 8% compared to 14% for all SNVs, again demonstrating strong selection against these mutations (Fig. 2f). There are many well-studied pathogenic mitochondrial SNVs that cause mitochondrial disorders. Of the 91 confirmed pathogenic SNV mutations on Mitomap[16], 60 unique variants are found in this dataset with 1 variant assigned to two genes, *MT-CO1* and tRNA serine 1 (*MT-TS1*) (Supplementary Data 1).

We have additionally identified 2 large-fragment deletions (314_955del and 8482_13446del) in the UKB participants. Of the 194,871 individuals evaluated, 314_955del was present in 72.5% (n = 141,420) and 8482_13446del in 0.05% (n = 89). The VAF for both deletions were

**Table 1 | Number of heteroplasmic SNVs per participant**

| Heteroplasmy count | Number of participants (%) |
|---|---|
| 0 | 135,457 (69.5%) |
| 1 | 47,008 (24.1%) |
| 2 | 10,273 (5.3%) |
| 3 | 1785 (0.92%) |
| 4 | 280 (0.14%) |
| 5 | 68 (0.03%) |

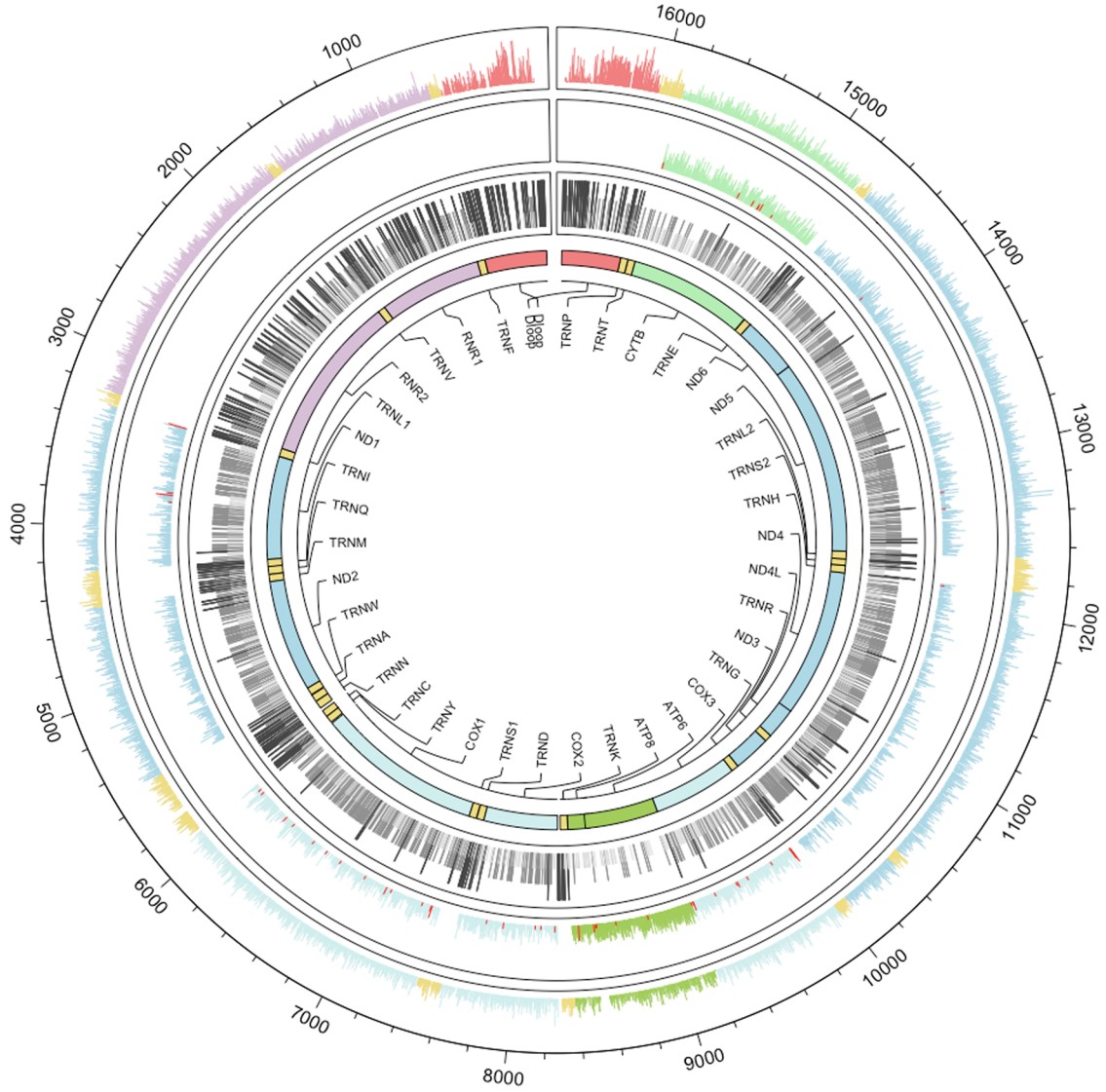

**Fig. 1 | Single nucleotide variants (SNVs) and invariant sites across mitochondrial DNA.** Circos plot tracks from outer (Track 1) to inner (Track 4): Track 1 is all synonymous and non-coding (tRNA, rRNA and D-loop) heteroplasmic sites. Track 2 is all nonsynonymous and nonsense heteroplasmic SNVs, where nonsense heteroplasmic SNVs are colored in red. The Y-axis of Tracks 1 and 2 is the log(number of participants with a heteroplasmy + 1), scaled from 0 to 9. Track 3 is positions with no heteroplasmy. Three or more adjacent null positions are colored light gray, 2 adjacent positions are colored medium gray, and singlets are colored dark gray. The height of Track 3 bars is scaled by color, light gray is the lowest, followed by medium gray, then dark gray. Innermost track (Track 4) is gene annotations. Genes are colored similarly by complex or by gene type. Source data are provided as a Source Data file.

low, with the median (range) of 0.24% (0.08–3.88%) for 314_955del and 0.12% (0.10–0.28%) for 8482_13446del.

## Heritability of heteroplasmic variants

Prior studies have demonstrated that a substantial proportion of heteroplasmies observed in any given individual are maternally inherited[6,17,18]. To directly assess whether UKB participants showed

## Table 2 | Frequency of heteroplasmic SNVs in the population

| Variant allele counts | Number of variants |
|---|---|
| Singletons | 4257 |
| 2–10 | 6040 |
| 11–100 | 1258 |
| 101–1000 | 43 |
| 1000+ | 4 |

This table summarizes the number of times that each SNV appears in the study population. For example, there were 4257 heteroplasmic SNVs that appeared in only one participant.

high levels of inherited heteroplasmies, we identified monozygotic twins (28 pairs), mother/child pairs (760 pairs), and full sibling pairs (3657 pairs), among the 194,871 participants. Of the 12 unique variants found in monozygotic twins, 58.3.% ($n = 7$) were shared between the twins, of the 352 variants found in mother/child pairs, 29.8% ($n = 105$) were shared between the mother and child pairs, and of the 1372 variants found in full sibling pairs, 29.4% ($n = 403$) were shared between the siblings (Supplementary Table 1). Thus, our findings were consistent with a previous study that approximately 30% of heteroplasmies observed are maternally inherited[6].

## Association with all-cause mortality

Given that ~70% of the observed mitochondrial heteroplasmic mutations are likely to be somatic mutations, we would expect that heteroplasmic mutations would increase with age, as has been previously shown[9,17], as well as with exposure to mutagenic chemicals such as tobacco. Indeed, we observed a significant increase in heteroplasmic SNVs both as a function of age and smoking status (Fig. 3). We also

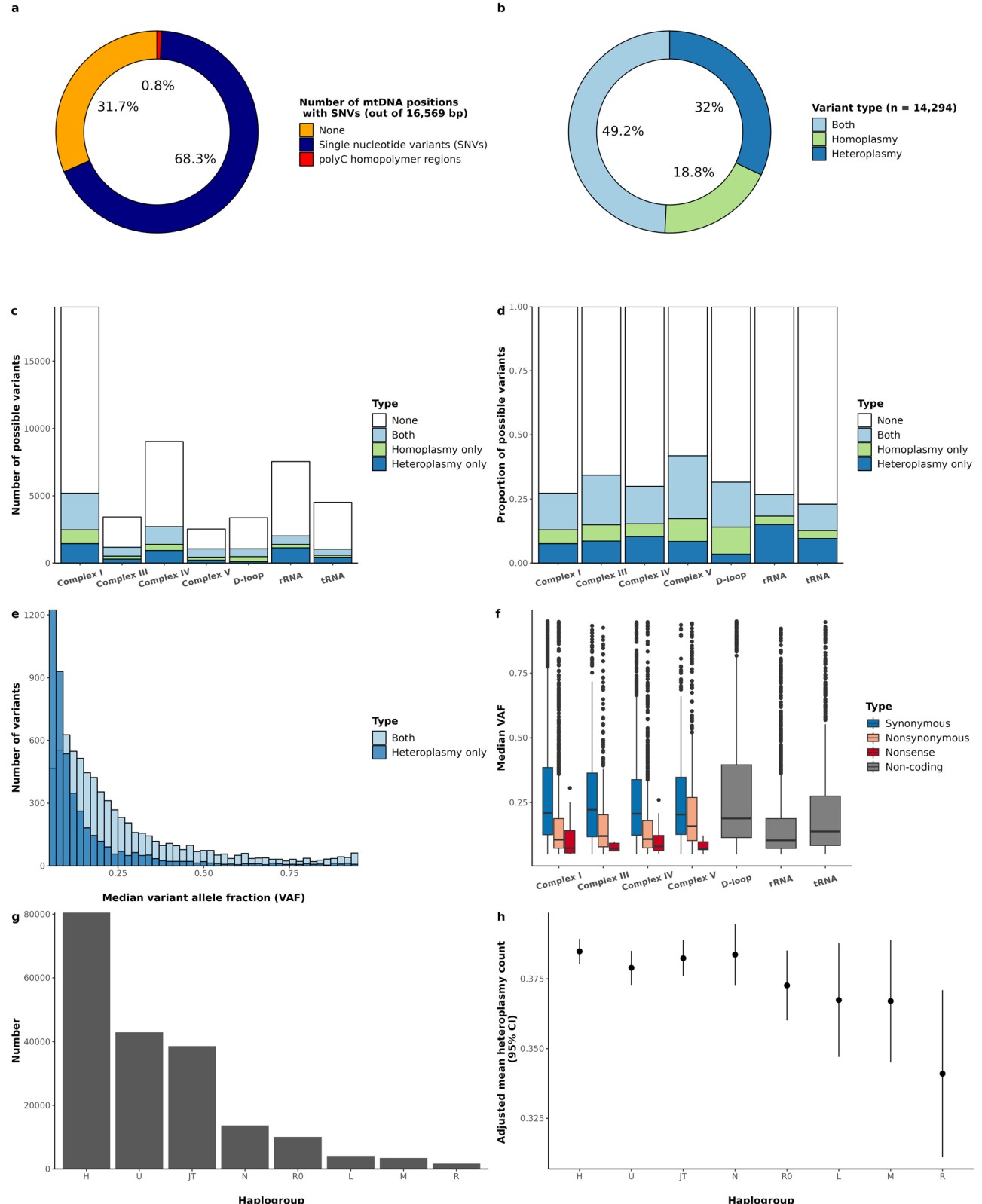

observed a significant interaction between age and smoking status ($P = 7.1 \times 10^{-4}$), likely reflecting the cumulative impact of long-term tobacco use. We did not see a significant association between sex and heteroplasmy count after adjusting for age and smoking status.

To determine the impact of mitochondrial heteroplasmy on all-cause mortality, we ran Cox proportional hazards models stratified by assessment center and adjusted for age, sex, and smoking status. We

observed a 1.50-fold (95% confidence interval [CI] 1.14, 1.98) increased risk of all-cause mortality in individuals with 4 or more heteroplasmies compared to those without any heteroplasmy (Fig. 4). This analysis, however, assumes 1) that all heteroplasmies have the same association (effect estimate) with mortality, and 2) it does not directly test the hypothesis that mitochondrial heteroplasmy is causal for increased mortality risk. An alternative hypothesis could be that heteroplasmy

**Fig. 2 | Overview of mitochondrial SNV distributions. a** Proportion of mitochondrial DNA positions with SNVs. **b** Proportion of SNVs that are heteroplasmic, homoplasmic, or both. **c** Number of possible SNVs that are heteroplasmic, homoplasmic, or both, grouped by protein complex or genic context. **d** Bar chart of the proportion of possible SNVs that are heteroplasmic, homoplasmic, or both grouped by protein complex or genic context scaled to 1. **e** Histogram of the median variant allele fraction (VAF) for variants seen as only a heteroplasmy or both a hetero- and homoplasmy among individuals with at least one heteroplasmic SNV (*n* = 59,414). The two histograms are overlaid. **f** Boxplot of the median VAF by type of mutation. The center line indicates the median, the box limits the lower and upper quartiles, the whiskers the 1.5× interquartile range, and the points outliers. **g** Number of participants in each haplogroup (*n* = 194,871). Haplogroups were grouped by phylogenetic similarity into the following: L is L0-L6; M is C, D, E, G, M, Q, Z; N is A, I, N, S, W, X Y; R is B, F, P, R; R0 is R0, HV, V; U is U, K; JT is JT, T; H is H only. **h** Mean heteroplasmy count (95% confidence intervals) by haplogroup adjusted for age, sex, center, and smoking status. Source data are provided as a Source Data file.

## Table 3 | Characteristics of heteroplasmic SNVs by gene type/region

|  | Length (bp) | Ti/Tv | dN/dS | # Nonsense mutation sites (nonsense SNVs/ length *1000) | # Nonsense mutation alleles | Mean MLC score (SE) |
| --- | --- | --- | --- | --- | --- | --- |
| Complex I | 6349 | 26.8 | 0.87 | 8 (1.3) | 9 | 0.23 (0.0048) |
| Complex III | 1141 | 25.5 | 1.32 | 6 (5.3) | 11 | 0.16 (0.0066) |
| Complex IV | 3010 | 24.4 | 1.20 | 28 (9.3) | 57 | 0.29 (0.0070) |
| Complex V | 842 | 19.9 | 1.92 | 7 (8.3) | 8 | 0.06 (0.0040) |
| D-loop | 1122 | 41.2 | – | – | – | 0.09 (0.0047) |
| tRNA | 1505 | 26.9 | – | – | – | 0.43 (0.0072) |
| rRNA | 2513 | 25.2 | – | – | – | 0.55 (0.0057) |
| Unassigned | 87 | 35.1 | – | – | – | 0.28 (0.0300) |
| Overall | 16,569 | 28.7 | 1.08 | 49 (4.3) | 85 | 0.28 (0.0029) |

Columns are defined as follows: length is the total number of bases by region or gene type. Protein-coding genes are grouped by oxidative phosphorylation (OXPHOS) complexes. The transition-to-transversion ratio (Ti/Tv) is calculated as the number of SNVs with transition changes (A <-> G or T <-> C) divided by the number of SNVs with transversion changes. Nonsynonymous-to-synonymous ratio (dN/dS) is calculated as the number of nonsynonymous mutations divided by synonymous mutations for all protein-coding SNVs. Number of nonsense mutation sites is the sum of all SNVs that cause a stop codon grouped by genes that make up OXPHOS complexes. Number of nonsense mutation alleles is the total number of nonsense mutation alleles present in each OXPHOS complex. Mitochondrial local constraint (MLC) score is calculated from SNVs observed in each gene type or region. The mean and the standard error (SE) of MLC score are reported.

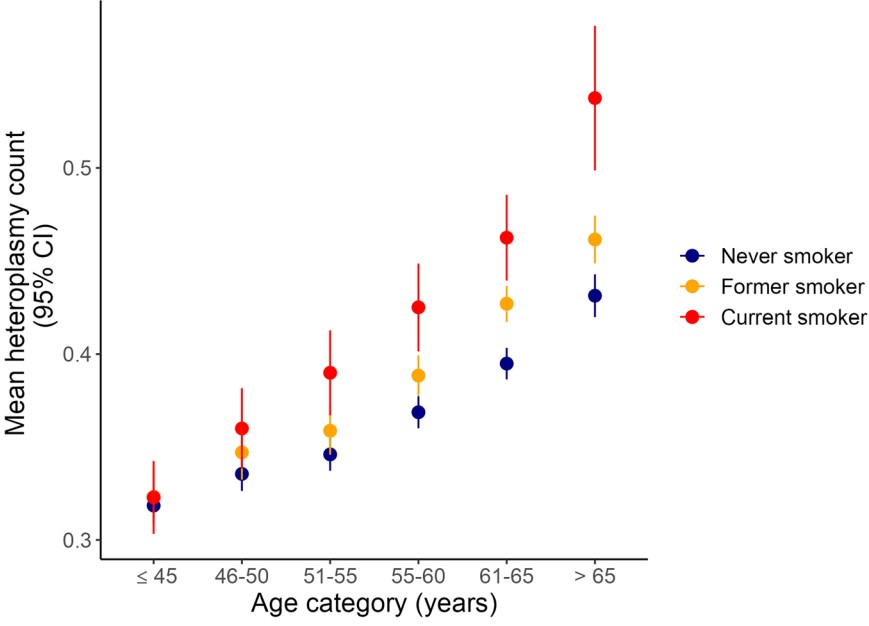

**Fig. 3 | Mean heteroplasmic count by age category and smoking status in the UK Biobank (*n* = 194,871).** Error bars are 95% confidence intervals. Source data are provided as a Source Data file.

count is a non-causal biomarker for some other disease process or environmental exposure, and under this non-causal model, we would predict that the specific functional impact of the heteroplasmic variant would not impact its association with mortality. To distinguish between the functional impact from the number of heteroplasmies, we incorporated functional annotations to test whether functional heteroplasmies are driving the association between the number of heteroplasmies and risk of mortality. While we see no significant association for those who carry a synonymous mutation (aHR 1.03, 95% CI 0.99, 1.07), we see a significant increased risk for those carrying a nonsynonymous mutation (aHR 1.10, 95% CI 1.05, 1.17) (Fig. 4). The highest mortality risk was observed in participants harboring a nonsense mutation, which introduces a stop codon and therefore, directly decrease protein function. The adjusted hazard ratio (aHR) for all-cause mortality was 1.77 (95% CI 1.07, 2.94) and further adjusting for heteroplasmy count marginally attenuated the association (aHR 1.65;

**a**

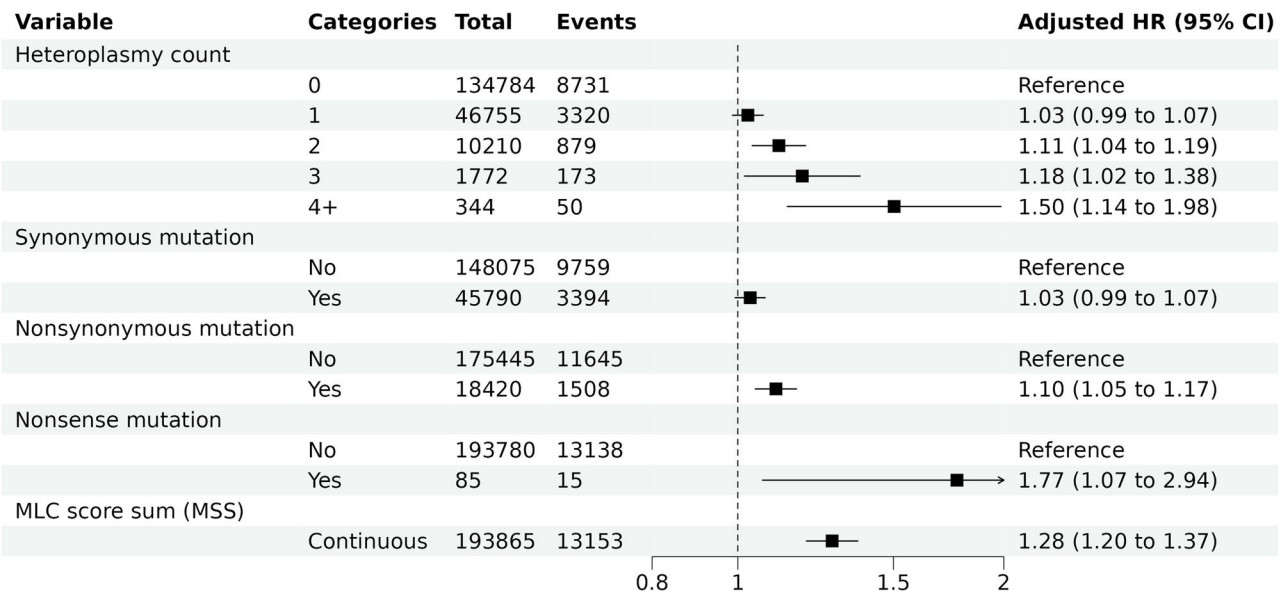

| Variable | Categories | Total | Events | | Adjusted HR (95% CI) |
|---|---|---|---|---|---|
| Heteroplasmy count | | | | | |
| | 0 | 134784 | 8731 | | Reference |
| | 1 | 46755 | 3320 | | 1.03 (0.99 to 1.07) |
| | 2 | 10210 | 879 | | 1.11 (1.04 to 1.19) |
| | 3 | 1772 | 173 | | 1.18 (1.02 to 1.38) |
| | 4+ | 344 | 50 | | 1.50 (1.14 to 1.98) |
| Synonymous mutation | | | | | |
| | No | 148075 | 9759 | | Reference |
| | Yes | 45790 | 3394 | | 1.03 (0.99 to 1.07) |
| Nonsynonymous mutation | | | | | |
| | No | 175445 | 11645 | | Reference |
| | Yes | 18420 | 1508 | | 1.10 (1.05 to 1.17) |
| Nonsense mutation | | | | | |
| | No | 193780 | 13138 | | Reference |
| | Yes | 85 | 15 | | 1.77 (1.07 to 2.94) |
| MLC score sum (MSS) | | | | | |
| | Continuous | 193865 | 13153 | | 1.28 (1.20 to 1.37) |

**b**

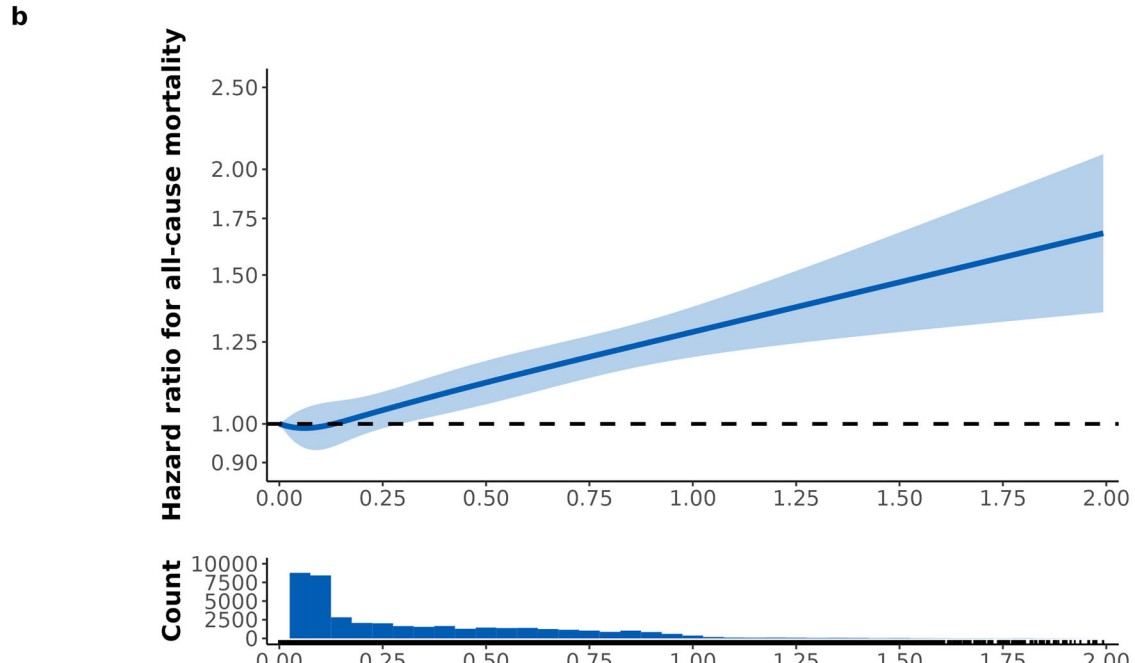

**Fig. 4 | Hazard ratios (95% confidence intervals) for all-cause mortality by heteroplasmy and mutation type. a** Hazard ratios (HR) for the association of all-cause mortality by heteroplasmy count, presence of synonymous mutation, non-synonymous mutation, nonsense mutation, and MLC score sum (MSS) were estimated from separately Cox proportional hazards models stratified by assessment center and adjusted for age, sex, and smoking status. The numbers of participants and events reflects the number included in the regression analysis. **b** Adjusted dose-response relationship between MSS and all-cause mortality and histogram of MSS among participants with at least 1 heteroplasmy count. Hazard ratios (95% confidence intervals) were derived from Cox proportional hazards models that include MSS as restricted cubic splines with 4 degrees of freedom and were stratified by assessment center and adjusted for age, sex, smoking status, alcohol intake, body mass index, white blood cell count, and haplogroup. 58 participants with MSS greater than 2 were excluded from the plot. Source data are provided as a Source Data file.

95% CI 0.99, 2.74; Fig. 4a, Supplementary Fig. 4a), suggesting an association of nonsense mutation and all-cause mortality independent of the number of heteroplasmies.

Second, we incorporated functional annotation into our models to test whether functional heteroplasmies drive the association between the number of heteroplasmies and risk of mortality. We used the mitochondrial local constraint (MLC) score[19], a recently developed annotation metric based on local constraint, which quantifies the local tolerance to base or amino acid substitutions for each base pair across the entire mtDNA genome. The MLC score assigns each heteroplasmy a score ranging from 0 to 1, with 1 being the most constrained SNV, and therefore, likely to have the most deleterious impact when mutated. The MLC score addresses the limitations in the algorithm currently recommended by the ACMG/AMP guidelines[20] for mtDNA variants by

incorporating variants in non-coding regions, tRNA, and rRNA, as well as non-missense mutations. We randomly selected one heteroplasmic SNV from each participant, and then tested whether the MLC score had a significant association with all-cause mortality. MLC score was significantly associated with mortality (adjusted HR comparing MLC score of 1 to 0 was 1.34; 95% CI 1.21, 1.49; $P = 5.4 \times 10^{-8}$; Supplementary Fig. 4b), while continuing to adjust for center (stratification factor), age, sex, smoking status, and the number of heteroplasmies. Consistent with lower dN/dS ratio, we observed higher MLC scores in Complex I and IV, compared to Complexes III (Bonferroni-corrected $P$ values $1.5 \times 10^{-16}$ and $4.1 \times 10^{-35}$) and V (Bonferroni-corrected $P$ values $5.9 \times 10^{-77}$ and $9.5 \times 10^{-105}$), suggesting higher selection constraint (Table 3). To capture the impact of multiple heteroplasmies, we generated an MLC score sum (MSS) by summing all MLC scores within a given individual. The association between heteroplasmy count and all-cause mortality was no longer significant after adjusting for MSS, while the risk of mortality increased with a higher MSS (aHR for 1-unit increase in MSS 1.28; 95% CI 1.20, 1.37; Supplementary Fig. 5). Furthermore, incorporating the MSS in addition to the presence of non-synonymous mutations largely explained the effect of heteroplasmy count and nonsynonymous mutation on all-cause mortality (Supplementary Fig. 5). In addition, we evaluated whether VAF is associated with all-cause mortality and did not observe any association between VAF and mortality (Supplementary Table 2), suggesting that MSS is more relevant for disease risk than VAF. Thus, for all downstream analyses, we used the MSS as a metric that captures the cumulative mutational burden of mitochondrial heteroplasmy. After additionally adjusting for alcohol intake, body mass index (BMI), white blood cell (WBC) counts, and haplogroup, MSS was associated with all-cause mortality in a dose-response manner (Fig. 4b; $P$ for non-linearity = 0.55) and a 1-unit increase in MSS was associated with a 28% (aHR 1.28; 95% CI 1.20, 1.37) increase in the risk of mortality. There were no significant differences by sex ($P$ for interaction = 0.6).

We have further validated our findings in additional cohorts using WGS from the TOPMed consortium (Atherosclerosis Risk in Communities [ARIC], Multi-Ethnic Study of Atherosclerosis [MESA], Framingham Heart Study [FHS], and Women's Health Initiative [WHI]), comprising a total of 31,408 participants and 14,335 events. We performed meta-analysis for the associations of heteroplasmy count, MSS, and overall mortality adjusting for age (restricted cubic splines with 4 degrees of freedom), sex, smoking status, and center. As seen in the UKB, while we observed smaller effect sizes for MSS, both heteroplasmy count and MSS were associated with overall mortality, with hazard ratios of 1.04 (95% CI 1.02, 1.06), and 1.12 (95% CI 1.06, 1.19), respectively (Supplementary Fig. 6a, b). Consistent with the UKB analyses, when both heteroplasmy count and MSS were included in the same model, the effect size of the two metrics were both attenuated (Supplementary Fig. 6c).

### Associations with all-cause mortality by race

When stratified by race, in the UK Biobank, both heteroplamsy and MSS were associated with overall mortality in those self-identified as White. In Black individuals, on the other hand, there was a positive trend between heteroplasmy count and overall mortality (HR 1.15; 95% CI 0.91, 1.44) but no association between MSS and overall mortality (HR 1.01; 95% CI 0.48, 2.11). When both heteroplasmy count and MSS were included in the model, MSS was positively associated with overall mortality in all race/ethnicity groups except in Black participants (Supplementary Fig. 7). However, given the small sample sizes, there was no statistical difference between the race/ethnicity groups for any of the three models.

To validate our findings and to increase sample size, we repeated the analysis including race-stratified results from the 4 additional TOPMed cohorts. In the pooled analysis, MSS was positively associated with overall mortality in Whites but not in Blacks (Supplementary

Fig. 8). The number of Hispanics and Asians were small in these cohorts to provide precise estimates and, thus, need to be validated with a larger sample size.

### Association with cause-specific mortality

We further investigated if the association with increased mortality was driven by a specific cause of death. When categorizing participants by cause of death using the International Classification of Diseases 10th Revision (ICD-10), MSS was nominally associated with an increased risk of death due to neoplasms (C00–D48; FDR-corrected $P$ value $2.1 \times 10^{-10}$), digestive disorders (K20–K93; FDR-corrected $P$ value 0.08), and external causes (V01–Y89; FDR-corrected $P$ value $1.6 \times 10^{-3}$) (Fig. 5a). There were no significant differences by sex. Additional adjustment for neutrophil, lymphocyte, and platelet counts did not significantly change the results. Given that the generic category malignant neoplasms ("cancer") includes many diseases from different organ systems, we further separated solid and hematologic cancers by type (Fig. 5b). The MSS was nominally associated with mortality due to lung cancer (aHR 1.56; 95% CI 1.27, 1.91; FDR-corrected $P$ value $7.8 \times 10^{-5}$), breast cancer (aHR 1.51; 95% CI 1.06, 2.15; FDR-corrected $P$ value 0.06), lymphoma (aHR 2.01; 95% CI 1.34, 3.01; FDR-corrected $P$ value $2.5 \times 10^{-3}$), and leukemia (aHR 4.97; 95% CI 3.87, 6.39; FDR-corrected $P$ value $1.6 \times 10^{-15}$). Of the external causes of death, the MSS was associated with a higher risk of death due to self-harm (aHR 2.47; 95% CI 1.35, 4.52) but was not associated with death due to accidents (aHR 0.94; 95% CI 0.13, 6.78). The associations did not change materially when we restricted the analyses to genetically unrelated participants, to individuals without any known mitochondrial disorder, to individuals without any of the 60 identified pathogenic mitochondrial SNVs, or to individuals without any heteroplasmies in regions that could be confounded by NUMTs. In addition, the associations of MSS with all-cause mortality and disease-specific mortality did not meaningfully change when we applied different heteroplasmy thresholds (3% and 10%), a mtDNA-CN threshold (mtDNA-CN > 60), and VAF 10% and mtDNA-CN > 60 simultaneously (Supplementary Fig. 9).

Clonal hematopoiesis of indeterminate potential (CHIP) is a key predictor for mortality and for blood cancers. Therefore, we generated CHIP calls in the UK Biobank as previously described[21] and accounted for the presence of CHIP in the associations of heteroplasmy count and MSS with overall mortality and incidence of hematologic cancer, lymphoma, and leukemia. Even after adjusting for CHIP, the results did not change materially, suggesting that the association of mitochondrial DNA heteroplasmy with mortality and blood cancers is independent of CHIP (Supplementary Table 3).

### Cancer prevalence, incidence, and mortality

The association with cancer-specific mortality could arise due to a combination of factors. First, MSS could be a biomarker for cancer presence (prevalence of disease), second, it could be associated with the development of cancer (incidence of disease), or third, it could be associated with survival in the presence of disease. We therefore evaluated the associations of MSS with prevalent and incident cancer, as well as survival in those with cancer, for each of the 4 significantly associated cancers (lung, breast, lymphoma, and leukemia) to better distinguish between these potential mechanisms. MSS was associated with both prevalent and incident cases of lung cancer, lymphoma, and leukemia (Fig. 6a, b). For incident lung cancer, the association was slightly attenuated after including smoking duration in the model with a HR 1.28 (1.03, 1.59). For breast cancer, MSS was associated with prevalent breast cancer (adjusted prevalence ratio [aPR] 1.17; 95% CI 1.01, 1.34; FDR-corrected $P$ value 0.04), whereas there was no association with incident disease (aHR 0.96; 95% CI 0.81, 1.13; FDR-corrected $P$ value 0.66). The findings suggest a role for MSS as a biomarker of all these cancers and, potentially, the involvement of mitochondrial dysfunction in the development of lung cancer, lymphoma, and leukemia.

**a**

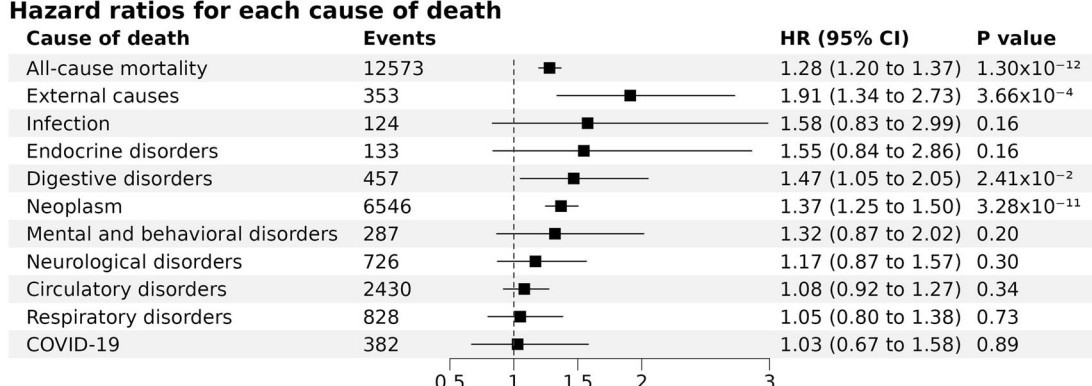

**b**

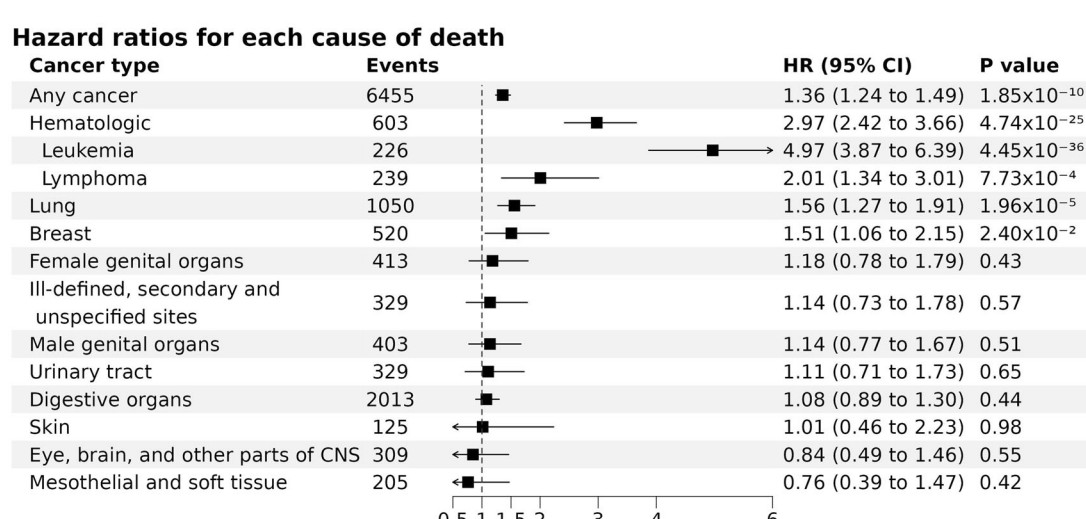

**Fig. 5 | Hazard ratios for each cause of death per 1-unit increase in MLC score sum (MSS).** Hazard ratios for **a** each specific cause of death, and **b** cancer-specific mortality, were estimated using Cox proportional hazards models stratified by center and adjusted for age, sex, smoking status, alcohol intake, body mass index, white blood cell count, and haplogroup. FDR-corrected *P* values accounting for multiple testing are reported in the main text. Uncorrected *P* values are presented in the plot. Cases of death in fewer than 100 participants (deaths due to benign diseases of the blood [*n* = 26] or genitourinary disorders [*n* = 83] in **a** and deaths due to cancers of lip, oral cavity, and pharynx [*n* = 91] or thyroid and other endocrine glands [*n* = 25] in **b**) are not included in the plot. Neoplasm in **a** included both benign and malignant neoplasms. Hematologic cancers in **b** included cancers of the lymphoid, hematopoietic, and related tissues. Lung cancers in **b** included cancers of the respiratory and intrathoracic organs. All statistical significance was based on two-sided tests. Abbreviations: CNS, central nervous system. The number of events reflect the number of events in participants included in the regression analysis. Source data are provided as a Source Data file.

We further evaluated the risk of mortality from each cancer among those who either had prevalent cancer at the time of UKB visit or had developed cancer (incident cancer) after visit. For all 4 cancers, higher MSS was more strongly associated with mortality among those with prevalent disease, though only significant for leukemia mortality (aHR 4.03; 95% CI 1.34, 12.11; *P* value 0.013; FDR-corrected *P* value 0.16) and breast cancer mortality (aHR 1.58; 95% CI 1.05, 2.40; *P* value 0.029; FDR-corrected *P* value 0.17) (Fig. 6c), suggesting a potential role of MSS as a prognostic marker of these cancers. The associations were similar when we restricted the analyses to genetically unrelated participants and to participants without extreme values of blood cell counts, which may be indicative of undiagnosed hematological cancers, and when we further adjusted for neutrophil, lymphocyte, and platelet counts.

**PheWAS and PHESANT results**
Finally, given the varied role that mitochondrial function plays in human health and disease, we examined whether the MSS was associated with a broad array of phenotypes. We first collapsed summary

ICD-10 diagnosis codes from UKB inpatient hospital records into 1618 broad phecodes capturing various disease categories, then tested association between the binary phecodes and MSS in a phenome-wide association study (PheWAS)[22] (Supplementary Data 2). As expected, there was an enrichment for hematological cancers among disease phecodes significantly associated with the MSS, as well as several lung-related phenotypes (lung cancer, bronchitis) (Fig. 7a). The MSS was further associated with hematopoietic disease phecodes, such as neutropenia, thrombocytopenia, and aplastic anemia. Notably, PheWAS revealed that the MSS was also associated with sepsis and septicemia phecodes, a finding that was significant even after excluding individuals with leukemia, who often experience sepsis during the course of their disease. A parallel phenome-wide association study was conducted using PHESANT[23], which additionally tests for associations with a broad array of phenotypes available in the UKB. The MSS was positively associated with several biomarkers of hematopoiesis including immature reticulocytes, platelets, and neutrophil counts, but inversely associated with eosinophil percentage (Supplementary Data 3). The MSS was also inversely associated with leukocyte T/S

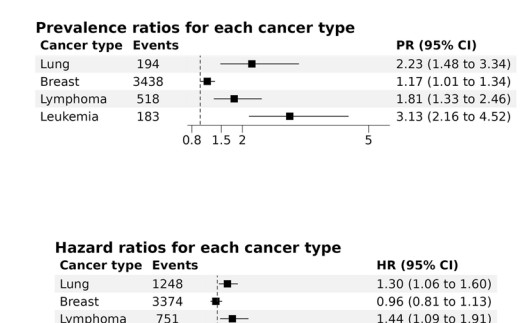

**a**

**Prevalence ratios for each cancer type**

| Cancer type | Events | | PR (95% CI) |
|---|---|---|---|
| Lung | 194 | | 2.23 (1.48 to 3.34) |
| Breast | 3438 | | 1.17 (1.01 to 1.34) |
| Lymphoma | 518 | | 1.81 (1.33 to 2.46) |
| Leukemia | 183 | | 3.13 (2.16 to 4.52) |

0.8  1.5  2          5

**b**

**Hazard ratios for each cancer type**

| Cancer type | Events | | HR (95% CI) |
|---|---|---|---|
| Lung | 1248 | | 1.30 (1.06 to 1.60) |
| Breast | 3374 | | 0.96 (0.81 to 1.13) |
| Lymphoma | 751 | | 1.44 (1.09 to 1.91) |
| Leukemia | 454 | | 3.47 (2.75 to 4.37) |

0.8  1.5  2          5

**c**

**Hazard ratios for mortality due to cancer type**

| Cancer type | Prevalent or incident | Total | Events | | HR (95% CI) |
|---|---|---|---|---|---|
| Lung | | | | | |
| | Any | 1635 | 891 | | 1.15 (0.93 to 1.41) |
| | Prevalent | 194 | 38 | | 1.47 (0.58 to 3.69) |
| | Incident | 1441 | 853 | | 1.14 (0.92 to 1.41) |
| Breast | | | | | |
| | Any | 6997 | 502 | | 1.38 (0.98 to 1.94) |
| | Prevalent | 3438 | 269 | | 1.58 (1.05 to 2.40) |
| | Incident | 3559 | 233 | | 1.13 (0.62 to 2.06) |
| Lymphoma | | | | | |
| | Any | 1355 | 204 | | 1.37 (0.84 to 2.23) |
| | Prevalent | 518 | 34 | | 1.61 (0.55 to 4.69) |
| | Incident | 837 | 170 | | 1.24 (0.70 to 2.20) |
| Leukemia | | | | | |
| | Any | 691 | 177 | | 1.26 (0.91 to 1.74) |
| | Prevalent | 183 | 16 | | 4.03 (1.34 to 12.11) |
| | Incident | 508 | 161 | | 1.18 (0.82 to 1.71) |

0.8  1.5  2          5

**Fig. 6 | The associations of MSS and cancer prevalence, incidence, and death. a** Prevalence ratios (PRs) and 95% confidence intervals (CIs) for the associations of MSS with prevalent lung cancer, breast cancer, lymphoma, and leukemia were estimated using marginally predicted prevalence from logistic regression models adjusted for age, sex, center, smoking status, alcohol intake, body mass index, white blood cell count, and haplogroup. **b** Hazard ratios (HRs) and 95% CI for the associations of MSS with incident lung cancer, breast cancer, lymphoma, and leukemia were estimated using Cox proportional hazards models adjusted as above. For incident cancer, we excluded participants with any history of cancer at the time of blood collection. **c** Sub-distribution HRs and 95% CIs for mortality due to lung cancer, breast cancer, lymphoma, and leukemia were estimated using the Fine and Gray method to account for competing events (mortality from other cancers and non-cancer causes). The analyses were restricted to participants who had prevalent cancer at the time of the UKB visit ("Prevalent"), new cases that developed during follow-up ("Incident"), or either prevalent or incident cancer ("Any") of the given type of cancer. For incident cases, individuals with prevalent cancer of the given type were excluded. The numbers of participants and events reflect the numbers included in the regression analyses. Source data are provided as a Source Data file.

ratios (adjusted, unadjusted, Z-adjusted), markers of relative telomere length. These results are consistent with prior work showing a positive correlation between another marker of mitochondrial function, mtDNA-CN, and telomere length[24,25], further suggesting a role for mitochondrial function in influencing telomere length. Finally, we tested association of the phecodes with identified mtDNA pathogenic variants (Supplementary Data 1), using carrier status coded as a synthetic allele to combine the seven selected pathogenic variants. PheWAS of the synthetic allele revealed a statistically significant association to primary/intrinsic cardiomyopathies (phecode 425.1, $P = 5.0 \times 10^{-5}$; number of cases = 1014, number of controls = 191,178, number of cases with a pathogenic variant = 10). Notably, m.3243 A > G made up a disproportionate frequency among carriers with phecode 425.1 (7/10 cases, 70%) compared to the population frequency (66/436 carriers of any pathogenic variant, 15.1%).

Based on the PHESANT results, we selected ICD-10 codes related to cancer and diseases of the blood, and stratified MSS by gene region/complex to see if different mitochondrial functional units were associated with different phenotypes. While high MSSs in Complex I and RRNA were broadly associated across the phenotypes, and the DLOOP and Complex V MSSs were largely not associated with any of the phenotypes, there were also distinct differences observed (Fig. 7b, Supplementary Fig. 10). High TRNA MSS was associated with chronic myeloproliferative disease, chronic monocytic leukemia, and myelodysplastic syndrome, and not associated with chronic lymphocytic leukemia (the strongest overall result). Complex III and IV MSSs were most significantly associated with chronic myeloproliferative disease, chronic lymphocytic leukemia, polycythaemia vera, and essential (hemorrhagic) thrombocythaemia (largely Complex IV).

## Discussion

Here we present a detailed examination of the impact of heteroplasmic mitochondrial SNVs, measured in blood, on all-cause mortality and a broad range of phenotypes in almost 200,000 individuals. We identified genes in Complex V as having the most nonsynonymous SNVs and Complex I as having the fewest after adjusting for gene size, along with the lowest dN/dS ratio, as has been observed in a previous large-scale study[5]. Likewise, we see a marked reduction in nonsense SNVs in Complex I genes, consistent with a strong negative selection against mutations that decrease Complex I function relative to the other

Complex genes. In a previous study evaluating mtDNA variation using 56,434 gnomAD samples[5], they identified a variant in 53% of the 16,569 mtDNA bases where nearly half of the unique variants were homoplasmic only and 37% were both homoplasmic and heteroplasmic at VAF 0.10–0.95. In our study, we identified a variant in 68.3% of the mtDNA nucleotides at VAF 0.05–0.95, where both homoplasmic and heteroplasmic variants (49.2%) and heteroplasmic only variants (32%) were more common than homoplasmic only variants. The differences may be due to different population characteristics as well as the different VAF cutoff used for defining a heteroplasmic variant. However, in both studies, most unique heteroplasmic variants were found in only one or two samples. In addition, in another study evaluating constraint in mitochondrial DNA in gnomAD samples, Complex III and V genes were more tolerant of missense mutation and Complex I and IV were less tolerant[19]. This is consistent with our findings of lower dN/dS ratios (fewer nonsynonymous mutations) and higher average mitochondrial local constraint (MLC) scores in Complex I and IV compared to Complex III and V.

We also identified over 66% of the known pathogenic mutations in our dataset. Some are observed as homoplasmic mutations like 14484 T > C, which was predicated to not cause LHON by Bolze et al. based on how frequently it occurred in their dataset (8 in 10,000) and the lack of individuals with the variant having an ICD-10 code associated with the disease[26]. Moreover, we found that mtDNA pathogenic variants, including m.3243 A > G, were associated with cardiomyopathies. Other studies have also identified m.3243 A > G to be associated with a number of phenotypes, including various cardiomyopathies and conduction disorders, and higher heteroplasmy levels of m.3243 A > G may be associated with severity and disease progression[27,28].

We found that the presence of heteroplasmy was associated with all-cause mortality and that a functional score based on local sequence constraint[19], MSS, was a better predictor variable than the number of heteroplasmies, implicating a potential role for mitochondrial heteroplasmy in all-cause mortality. These results were supported by the observation that, while synonymous mutations were not associated with increased mortality risk, nonsynonymous mutations were associated with increased mortality risk, with the presence of nonsense mutations associated with highest all-cause mortality risk. We also found MSS to be a better predictor of mortality than heteroplasmy

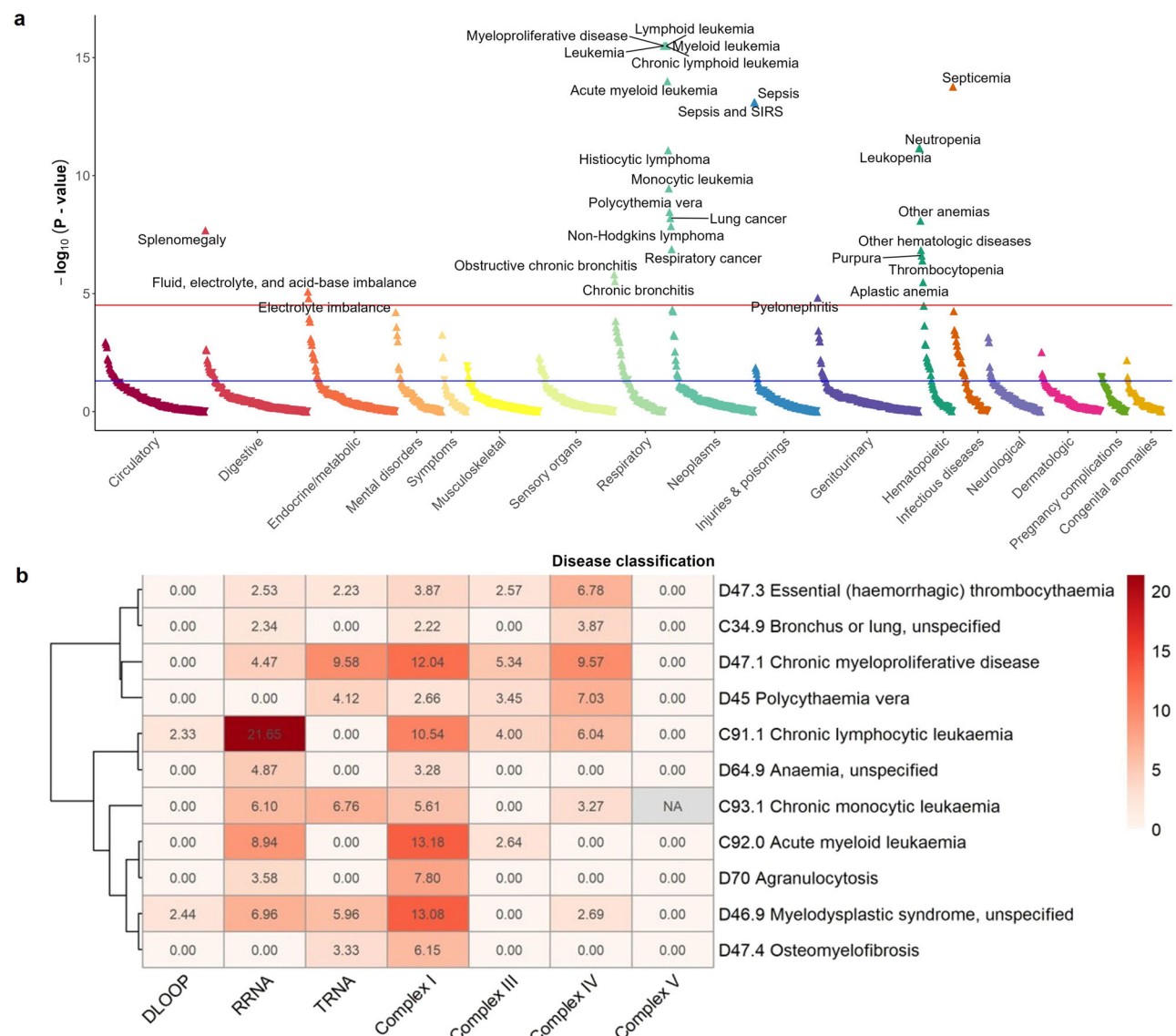

**Fig. 7 | Phenome-wide association study (PheWAS) of MSS. a** X axis indicates disease categories (colors) for phecodes with at least 10 cases, y axis indicates P value for association between phecode and MSS. Up/down arrows indicate positive/negative effect direction for association. The blue horizontal line indicates P value = 0.05. The red horizontal line indicates P value cutoff after Bonferroni correction $(3.1 \times 10^{-5})$. Phecodes with P values < $1.0 \times 10^{-15}$ are plotted at $-\log_{10}(P$ value)) = 15. **b** Heatmap displaying the significance ($-\log_{10}(P$ value)) of the association between MSS and ICD-10 codes for cancer ("Chapter II Neoplasms") and hematological diseases ("Chapter III Disease of the blood and blood-forming organs and certain disorders involving the immune mechanism") stratified by mtDNA region/complex. ICD-10 codes were selected if the significance with the overall MSS was $<1 \times 10^{-6}$, and clustered using Pearson's correlation. $-\log_{10}(P$ values) >2 (corresponding to P values > 0.01) were set to 0 for visualization. P values are not corrected for multiple comparison. All statistical significance was based on two-sided tests. Source data are provided as a Source Data file.

level, or VAF, which is in line with our observation that more deleterious mutations tend to have lower VAF, potentially because they are less tolerated. We, however, may have missed any effects that occur at lower VAF, as we have defined heteroplasmy at a threshold of 5%. We additionally replicated the analyses of heteroplasmy count, MSS, and all-cause mortality in a meta-analysis of 4 additional cohorts and found that, while both heteroplasmy count and MSS were associated with all-cause mortality, MSS was a better predictor. Additionally, when stratified by race, we observed an interesting finding in Black individuals where heteroplasmy count was a stronger risk factor for all-cause mortality with no statistically significant association between MSS and mortality. The different trends of mitochondrial heteroplasmy on all-cause mortality in Black participants may be due to potential differences in social and environmental factors that were not accounted for, differences in the performance of MLC score modeling by race (such as

differences in the location of variants or the constraints of each variant) and need to be further explored.

In addition to all-cause mortality, MSS was associated with an increased risk of specific causes of death, including neoplasms, digestive disorders, and external causes. Of the external causes of death, MSS was associated with a higher risk of death due to self-harm. In one case-control study comparing mtDNA-CN in suicide completers to controls, mtDNA-CN in peripheral blood was higher in suicide completers but lower in post-mortem dorsolateral prefrontal cortex after adjusting for age and sex, suggesting a possible role of mitochondrial function in self-harm[29]. Although psychiatric presentations can occur in primary mitochondrial disorders[30], further studies are needed to better understand the relationship between mitochondrial function and mental health, as well as to validate our findings.

Our most notable finding is the potential of using mitochondrial heteroplasmy as a biomarker of prevalence, incidence, or prognosis of certain cancers. For instance, we found that MSS was associated with the prevalence and incidence of lung cancer, lymphoma, and leukemia. In addition, high MSS was strongly associated with mortality due to leukemia and breast cancer in individuals who had a history of these cancers at the time of blood collection. These results present different models for how mitochondrial dysfunction, measured as the MSS, could increase the risk for cancer mortality. For breast cancer, MSS was not significantly associated with prevalence or incidence of disease, and only appears to influence breast cancer mortality. In contrast, the associations with lung cancer and lymphoma mortality appear to be largely driven by the association of MSS with both prevalence and incidence of disease, with no significant association with mortality observed among those with disease. Finally, for leukemia, MSS is associated with all three features, with extremely large effect sizes for prevalence, incidence, and mortality due to disease (all PR/HR > 3); however, there were relatively few deaths due to leukemia relapse in this dataset (n = 17). While we were limited to mtDNA heteroplasmy measured in blood and, thus, were not able to evaluate the association of MSS measured in additional tissues with mortality or cancer risk, various studies have found an excess of functional mtDNA heteroplasmies in many cancers, including lung and breast cancers tissues[31,32].

A recent comprehensive study from the International Cancer Genome/The Cancer Genome Atlas Pan-Cancer Analysis of Whole Genome (PCAWG) Consortium[32] looking at WGS from 2,658 cancers across 38 tumor types provides compelling evidence for enrichment of mtDNA truncating mutations in several cancer types, including kidney, colorectal, and thyroid cancers. Similarly, a study of 10,132 paired tumor/normal samples with whole exome sequencing from The Cancer Genome Atlas (TCGA) identified 4,381 mtDNA mutations and various levels of truncating mutations in several cancer types, including both hematologic and solid cancers[31]. These studies support the idea that mtDNA heteroplasmy detected in blood may be a viable biomarker for even some non-hematologic cancers. However, there have been a limited number of studies linking mitochondrial heteroplasmy in blood and hematological malignancies[33]. Intriguingly, while of marginal significance, one study demonstrated that patients with clinically refractive leukemia tended to have higher rates of amino acid-altering mutations[34]. Accordingly, regular screening for cancer-causing mtDNA mutations or SNVs at highly constrained sequences (high MLC score) could be developed as a prognostic marker for cancer patients and surveillance tool for cancer survivors.

There are a few limitations of our study. First, our analyses were limited to heteroplasmies with a VAF ≥ 5% to limit potential confounding by NUMTs, and we only analyzed SNVs at ≥30× depth. Regions with low coverage would appear to have no SNVs in our analysis, at any VAF. This is particularly evident in the 50 bp region downstream of the chrM start site where, even with the circularization approach to mapping in MitoHPC, we still had lower coverage and thus did not count SNVs identified in this region. Second, we only analyzed SNV mutations, as we do not yet have confidence in non-SNV calls (insertions/deletions). Thus, we are missing important sources of mitochondrial DNA variation that are likely to contribute to disease risk. Third, we did not validate non-recurrent variants by an independent sequencing method. Finally, we have chosen to focus on heteroplasmic variation given the reduced impact of selection on somatic mutation, but fully acknowledge that homoplasmic variation, as well the nuclear genome background on which both heteroplasmic and homoplasmic variants fall, may also have important impacts on disease risk.

The first disease-causing SNV of mtDNA was identified in 1988[35,36]. Now with next-generation sequencing scientists can query the full mitochondrial genomes of hundreds of thousands of individuals to investigate how mtDNA mutations contribute to common diseases. We found that mitochondrial heteroplasmic mutational burden, measured as the sum of a sequence-based local constraint score[19], is associated with all-cause and cancer-specific mortality. Moreover, this score has the potential to be used as a prognostic marker, identifying those at risk of both developing disease, as well as those who may benefit from more careful monitoring or intervention post disease onset.

## Methods

### UK Biobank WGS data
The UK Biobank was approved by the UK Biobank Research Ethics Committee and all participants provided written informed consent before participation. The UK Biobank is a large population-based prospective study of 500,000 participants aged between 40 and 69 years recruited across the United Kingdom from 2006 to 2010[10]. The UK Biobank collects extensive phenotypic and genotypic data on participants, which are used in our analysis. We included in our analysis 199,909 participants who underwent WGS of the DNA from the blood draw and consented to be in the study. The current study was approved by the Johns Hopkins Medicine Institutional Review Board.

### TOPMed WGS data
We analyzed WGS data from four population-based longitudinal cohort studies; Atherosclerosis Risk in Communities (ARIC) study, Multi-Ethnic Study of Atherosclerosis (MESA) study, Framingham Heart Study (FHS), and Women's Health Initiative (WHI). Datasets from the Trans-Omics for Precision Medicine (TOPMed) program are from freeze 8, and for ARIC, we had additional data from the Centers for Common Disease Genomics (CCDG) program. TOPMed studies provide WGS data at ~30× genomic coverage using Illumina next-generation sequencing technology, which must pass specific quality control metrics before being released for use by the scientific community. Additional information on TOPMed WGS data generation and processing methods can be found here: https://topmed.nhlbi.nih.gov/methods.

The ARIC study is a community-based, prospective cohort study focusing on risk factors for heart disease and stroke which recruited 15,792 individuals aged 45 to 64 years between 1987 and 1989 from 4 communities in the US (Forsyth County, NC; Jackson, MS; Minneapolis suburbs, MN; and Washington County, MD). All-cause mortality was ascertained through National Death Index database or in annual follow-up from baseline to 31 December 2019. DNA samples were collected from different visits and DNA for WGS were isolated from buffy coat using the Gentra Puregene Blood Kit (Qiagen). We used all available data with WGS information (n = 12,842) where 69.4% (n = 8926) was analyzed as part of the CCDG program and 30.6% (n = 3927) was analyzed as part of the TOPMed program. For the 11 individuals who were included in both programs, we randomly selected one as heteroplasmy count and MSS were identical for the two methods. We excluded participants (n = 133) who did not have information on visit of DNA collection (n = 83), had heteroplasmy count ≥6 (n = 38) or had mtDNA-CN ≤ 40 or missing values of mtDNA-CN (n = 13). We further excluded participants who or smoking status (n = 30). The final sample for analysis included 12,679 individuals (5664 men and 7015 women). In the study sample, the mean age was 57.4 (6.0) years, and 78.5% (n = 9953) and 21.5% (n = 2726) were self-identified as Whites and Blacks, respectively.

The MESA study is a community-based, prospective cohort study focusing on subclinical cardiovascular disease (CVD) and risk factors for CVD in diverse populations. The MESA study recruited 6,814 individuals aged 45 to 84 and free of prevalent clinical CVD between 2000 and 2002 from 6 communities in the US (Baltimore, MD; Chicago, IL; Forsyth County, NC; Los Angeles, CA; New York, NY; and St. Paul, MN). All-cause mortality was ascertained through annual follow-up surveys and tracking through the national death index. DNA samples were collected on visit 1 (baseline visit) and DNA was isolated from packed

cells using the Gentra Puregene Blood Kit. We used data from 4603 individuals with WGS information, informed consent, and follow-up information. We excluded participants who did not pass QC filtering ($n = 351$) due to a temporary change in method causing lower DNA extraction. We further excluded participants ($n = 16$) who had other sequencing issues ($n = 4$), heteroplasmy count $\geq 6$ ($n = 4$), or did not have information on smoking status ($n = 8$). There were no individuals with had mtDNA-CN $\leq 40$. The final sampled for analysis included 4236 individuals (2053 men and 2183 women). In the study sample, the mean age was 61.0 (9.9) years, and 41.2%, 23.2%, 22.4%, and 13.2% were self-identified as White, Black, Hispanic, and Asian, respectively.

The FHS study is a community-based, prospective cohort study to investigate the risk factors for cardiovascular disease. The recruitment of Original cohort was initiated in 1948 for individuals aged 30–62 ($n = 5209$) and was examined every two years. The Offspring cohort was recruited in 1971 and has been examined every 4–8 years. The Original cohort had 32 exams and the Offspring cohort had nine exams. The Third generation cohort and a small number of spouse individuals of the second generation were recruited between 2002 and 2005 and have had three examinations. All recruited individuals were White. All-cause mortality was ascertained through death certificates. DNA samples were collected at different visits and DNA for WGS were isolated from buffy coat using the in-house protocol following the standard DNA extraction and purification procedures. We used data from a total of 4196 participants whose WGS was done by TOPMed (first generation, $n = 376$; second generation, $n = 2218$; spouses of the second generation, $n = 95$; and third generation, $n = 1507$). Mitochondrial heteroplasmy was defined at VAF of 3%. We excluded participants ($n = 518$) who had heteroplasmy count $\geq 6$ ($n = 155$), mtDNA-CN $\leq 40$ ($n = 2$), or did not have information on smoking status ($n = 370$). The final sampled for analysis included 3678 individuals (1714 men and 1964 women) with the mean age of 58 (14) years.

The WHI is a prospective national health study to investigate and identify prevention strategies for major chronic diseases in women. It is composed of a clinical trials component (CT, $n = 68,132$), focused on the primary prevention of major chronic diseases in older women, and an observational component (OS, $n = 93,676$), focused on the identification of predictors of incident chronic disease and mortality. A total of 161,808 racially and ethnically diverse postmenopausal women of 50 to 79 years old were recruited between 1993 and 1998 from 40 clinical centers across the US. Of these individuals, WGS was done on 11,031 individuals who were selected for a nested case-control ancillary substudy of venous thromboembolism and stroke. All-cause mortality was ascertained by linkage through the National Death Index. DNA samples were collected at either baseline visit or the next available annual visit. DNA was isolated from peripheral leukocytes using one of the following extraction methods: phenol-chloroform extraction, salt precipitation, Qiagen Five Prime, or Qiagen Puregene/Bioserve. We exclude participants who had heteroplasmy count $\geq 6$ ($n = 7$). There were no individuals with had mtDNA-CN $\leq 40$. We further excluded participants ($n = 208$) who did not have information on time of death ($n = 34$), smoking status ($n = 153$), or race/ethnicity ($n = 21$). The final sampled for analysis included 10,816 women. In the study sample, the mean age was 68.6 (6.9) years, and 81.4% ($n = 8805$), 12.9% ($n = 1397$), 3.8% ($n = 304$), and 1.9% ($n = 201$) and were self-identified Whites, Blacks, Hispanics, and Asians respectively.

## MitoHPC on DNA nexus (heteroplasmy and mtDNA-CN)
All UK Biobank data was processed on the DNA Nexus server. MitoHPC uses GATK Mutect2[37,38] for variant identification. WGS CRAM files were used for variant calling, haplogroup identification, and contamination checks. We implemented a heteroplasmy allele frequency of 5%, meaning that variant alleles at a frequency 5-95% in an individual are counted as heteroplasmic. Alleles less than 5% or greater than 95% are counted as homoplasmic. We used SAMtools[39] to generate read count

and coverage information for mtDNA-CN calculations, using the command 'samtools idxstats'. Documentation on how to run MitoHPC on DNA Nexus server is available here: https://github.com/ArkingLab/MitoHPC/blob/main/docs/DNAnexus_CLOUD.md.

## MitoHPC on TOPMed (heteroplasmy and mtDNA-CN)
All TOPMed WGS Mitochondrial and NUMT reads were first extracted on Google Cloud using the command 'samtools view -b sample.bam chrM chr1:629084-634672 chr17:22521208-22521639'. The extracted data was then transferred to a local cluster and MitoHPC and SAMtools were used to generate heteroplasmy and mtDNA-CN information, following a similar approach to that used in the UK Biobank cohort. The Documentation on extracting Mitochondrial and NUMT reads from Google Cloud is available here: https://github.com/ArkingLab/MitoHPC/blob/main/docs/GOOGLE_CLOUD.md.

## Mitochondrial local constraint (MLC) score
The mitochondrial local constraint (MLC) score is a measure of local tolerance to base or amino acid substitutions. It was calculated for every possible mtDNA single nucleotide variant (SNV) by applying a sliding window method[19]. In brief, starting from position m.1 a window of length k is drawn and the observed:expected (oe) ratio of substitutions in gnomAD within the window and its 90% confidence interval (CI) is calculated. The window start position is then moved by 1 bp, and this process is repeated until a start position of m.16569 is reached. For positions in protein genes only missense variants are included in calculations to restrict to amino acid substitutions, while for all other positions all base substitutions are used. The mean oe ratio 90% CI upper bound fraction (OEUF) of all k windows overlapping each position is computed, and percentile ranked to achieve a score range of 0–1 where 1 is most constrained position and 0 is least constrained. A MLC score is obtained for every mtDNA SNV as follows: non-coding, RNA and missense variants are assigned their positional score, and non-missense in protein genes are assigned scores based on the OEUF value of the variant class with synonymous, stop gain, and start/stop lost being assigned scores of 0.0, 1.0, and 0.70 respectively. Variants with higher scores are predicted to be more deleterious. To capture the impact of multiple heteroplasmies, an MLC score sum (MSS) was generated by summing all MLC scores within a given individual.

## Sample QC and variant filtering
We ran 200,000 WGS samples in the UK Biobank (UKB) database through MitoHPC. Of those 199,919 samples had outputs from MitoHPC variant calling and of those we calculated mtDNA copy number (mtDNA-CN) in 199,909 samples (Supplementary Fig. 1). The median nuclear genomic coverage for WGS samples in the UKB is 35× and the median mtDNA coverage is 1113x. This allowed us to confidently call heteroplasmy with variant allele fractions (VAF) as low as 5%.

MitoHPC outputs various metrics for assessing sample quality, allowing us to remove low quality samples prior to analysis. MitoHPC uses Mutect2 for variant calling and outputs variant annotations that we used to filter for high quality variants. We excluded variants with read depth <300 and those flagged as base quality, strandedness, slippage, weak evidence, germline, position flags in the FILTER column of the VCF. We further excluded heteroplasmic variants at polyC homopolymer regions on the mitochondrial chromosome and excluded INDELs. INDELs are often found at homopolymer regions and due to the nature of these regions, are challenging to accurately call heteroplasmies[9]. Nuclear-encoded mitochondrial sequences (NUMTs) can contribute to false positive mtDNA heteroplasmy calls. However, the nuclear genome coverage is ~31× lower than the mitochondrial genome coverage and by implementing at mtDNA-CN cutoff, we reduce the influence of NUMTs on our data. We excluded samples based on a few criteria: potential mitochondrial contamination, 2 or

more variants belonging to a different mitochondrial haplogroup, multiple variants predicted to be NUMTs, low minimum base coverage, and low mean base coverage (Supplementary Fig. 2) which resulted in 2501 participants being excluded. Since mitochondrial heteroplasmy has previously been shown to be affected by low mtDNA-CN[5], we removed participants with mtDNA-CN less than 40 (*n* = 3580; Supplementary Fig. 1 and Supplementary Fig. 2b, c). Some samples met multiple exclusion criteria. We found that 358 participants had a heteroplasmic count above 5, with 175 of them identified as contaminated. We removed the remaining high heteroplasmic samples as these appeared to be outliers in our dataset with potentially unidentified contamination. Finally, we were left with 194,871 participants to use for downstream analysis. We plotted the distribution of mtDNA-CN and heteroplasmic count for each participant, to visualize how these cutoffs affect the data (Supplementary Fig. 2d).

## Clonal hematopoiesis of indeterminate potential (CHIP) calls

We identified the presence of clonal hematopoiesis of indeterminate potential (CHIP) following a published protocol using the UK Biobank data[21] which includes a list of 43 nuclear variants identified as CHIP genes. Variants were included if (1) belonged to a list of recurring hotspot mutations associated with CH and myeloid cancer; (2) have been reported as somatic mutations in hematological cancers at least seven times in the Catalog of Somatic Mutations in Cancer (COSMIC); or (3) met the inclusion criteria of a predefined list of putative CH variants. Germline variants were included if (1) the number of cases in the cohort flagged as germline was lower than the ones flagged as PASS; and (2) at least one of the cases had a *P* < 0.001 for a one-sided exact binomial test, where the null hypothesis was that the number of alternative reads supporting the mutation was 50% of the total number of reads (95% for copy number equal to one), except for hotspot mutations. Variants were excluded if they were not present in COSMIC, nor in the list of hotspots that had a MAF equal to or higher than $5 \times 10^{-5}$ and either the mean VAF of all cases was higher than 0.2 or the maximum AF was lower than 0.1. Frameshift, nonsense and splice-site mutations not present in COSMIC or in the hotspot list were also excluded if for each variant none of the cases had a *P* < 0.001 for a one-sided exact binomial test.

## Deletions in mtDNA

Deletions in the mtDNA were identified by analyzing split read alignments. Split reads are reads from a unique region of the read that have two or more alignments to the reference with the same orientation. We identified the alignment end positions on chrM (1st alignment 3′ & 2nd alignment 5′) and counted their frequencies. A minimum frequency of 2 was used. We also confirmed that none of the split reads had a mate mapped to other chromosomes, which could be a NUMT indicator.

## Statistical analysis

Statistical differences in the distribution and characteristics of SNVs between complex/regions were tested using Chi-square tests for counts, logistic regression models for binary variables, and linear regression models for continuous variables.

To assess the heritability of heteroplasmic variants, we estimated genetic relatedness between participants in the UK Biobank using Kinship-based Inference for Gwas (KING) software[40] and further identified monozygotic twins, mother/child pairs, and full sibling pairs. We, then, calculated the number of unique variants identified in each of the pairs, and the number of unique variants that were common in the pairs.

The associations of age, smoking status, and heteroplasmy count were evaluated using a Poisson regression with an interaction between age using restricted cubic splines with 4 degrees of freedom and smoking status (never, former, or current). *P* for interaction was obtained by comparing a model with and without the interaction term using likelihood-ratio test.

The UK Biobank is linked to national death registries and provides information on date of death and cause of death. Primary cause of death was coded using the ICD-10 codes and classified into 12 categories (infection [A00–B00, L00–L08]; neoplasm [C00–D48]; benign disease of the blood [D50–D89]; endocrine disorders [E00–E90]; mental and behavioral disorders [F00–F89]; neurological disorders [G00–G99]; circulatory disorders [I05–I89]; respiratory disorders [J09–J99]; digestive disorders [K20–K93]; genitourinary disorders [N00–N98]; COVID-10 [U07]; and external causes [V01–Y89], Supplementary Table 4). Neoplasm-related mortality was further separated into benign neoplasm [D00–D48] and malignant neoplasm [C00–C97], which was further categorized by type of cancer.

The UK Biobank also provides information on cancer diagnosis by linkage to national cancer registries, including type of cancer, date of cancer diagnosis, and age at cancer diagnosis. The type of cancer is coded using ICD-9 or ICD-10 codes, which we used to categorize into 15 types of cancer by organ system (Supplementary Table 5). For malignant neoplasms, stated or presumed to be primary, of lymphoid, hematopoietic and related tissue ("hematologic cancers", ICD-10 C81–C96, ICD-9 200–208), we further identified individuals with lymphoma (ICD-10 C81–C86, ICD-9 200–202) or leukemia (ICD-10 C91–C95, ICD-9 204–208). ICD-10 codes from the death registry were also used to identify participants who were diagnosed with cancer. Skin cancer only included melanoma cases and breast cancer cases were restricted to women. We used date of cancer diagnosis (from national cancer registries) or date of death (from the death registry), whichever came first, as the date of cancer diagnosis in the analysis. Cancers diagnosed before the UKB visit were defined as prevalent cancers and cancers diagnosed after the UKB visit were defined as incident cancers.

Survival analyses for all-cause mortality and disease-specific mortality were performed using Cox proportional hazards model to estimate the hazard ratios (HRs) and their corresponding 95% CIs. Each participant was followed from the date of visit to the date of death or to 12 November 2021 (administrative censoring), whichever came first. For COVID-19, on the other hand, we allowed late entries where follow-up started from the first date of COVID-19 mortality on record. Models for evaluating the association between heteroplasmy count and all-cause mortality were stratified by assessment center and were adjusted for age using restricted cubic splines with 4 degrees of freedom, self-identified sex, and smoking status (never, former, or current). Models evaluating the association between VAF and all-cause mortality included the same set of covariates. However, because each variant exhibits different heteroplasmy VAF, complicating the assignment of a single VAF for each individual, we evaluated the association of VAF on all-cause mortality under three different scenarios: (1) VAF of a randomly selected heteroplasmic SNV in each individual; (2) VAF of the heteroplasmic SNV with the largest MLC score in each individual; and (3) VAF of variants in individuals who carry a single variant. We further adjusted for MSS and compared any change in the estimates before and after adjustment. Models evaluating the associations of MLC score sum (MSS) with all-cause mortality and disease-specific mortality were additionally adjusted for alcohol intake (never, former, or current), BMI (continuous), WBC counts (continuous), and haplogroup (L, M, N, R, R0, U, JT, and H). To account for potential confounding by differential WBC counts and platelets, we repeated the analysis further adjusting for neutrophil, lymphocyte, and platelet counts.

A priori to the study, the mitochondrial heteroplasmy threshold level of 5% was chosen based on read depth to maximize ability to detect heteroplasmies while limiting possible confounding due to NUMTs. However, to validate our findings across different levels of mitochondrial heteroplasmy thresholds, we repeated our analysis using 3% and 10% cutoffs. To further exclude the possibility of NUMT contamination, we repeated the analysis after excluding individuals (*n* = 752) with a heteroplasmy that mapped to a mtDNA region that may

be confounded by NUMTs and captured by the MitoHPC based on a database of 38 common NUMT sequences and 382 NUMT SNVs (MitoHPC/RefSeq/NUMT.vcf.gz). Additionally, we applied a different mtDNA-CN threshold (mtDNA-CN > 60) and repeated the analysis to also reduce the possibility of NUMT contamination. To evaluate the non-linear association between MLC score and all-cause mortality, MLC score was modeled using restricted cubic splines with 4 degrees of freedom.

In addition, we evaluated the risk of mortality from cancers of the lung, breast, lymphoma, and leukemia by MSS using a proportional sub-distribution hazards model[41] to account for competing events (mortality from other cancers and non-cancer deaths, separately). Individuals who developed cancer during the study period (incident cancer) were followed from the date of cancer diagnosis whereas individuals with prevalent cancer at the time of assessment visit were followed from the date of visit (late entry) to account for immortal person time. The competing risk models were adjusted for age (restricted cubic splines with 4 degrees of freedom), sex, assessment center, smoking status, alcohol intake, BMI, WBC counts, and haplogroup.

We further estimated the prevalence ratios (PRs) of the 4 types of cancers using logistic regression models. We used marginally predicted probabilities to calculate PRs associated with a 1-unit increase in MSS adjusted for age (restricted cubic splines with 4 degrees of freedom), sex, assessment center, smoking status, alcohol intake, BMI, WBC counts, and haplogroup. The corresponding 95% CIs were estimated using the delta method. For the same 4 types of cancers, HRs and their 95% CIs were estimated using Cox proportional hazards models stratified by assessment center and adjusted for the same covariates. Participants with any cancer diagnosis prior to date of visit were excluded from the analyses of incident cancers. Participants were followed from the date of visit to date of cancer diagnosis, date of death, or administrative censoring, whichever came first. All statistical significance was based on two-sided tests.

As the UKB includes genetically related participants, we additionally repeated the analyses restricting to genetically unrelated participants (as defined in the UK Biobank as the variable used.in.pca.calculation). Moreover, we further restricted the study population to participants without extreme values of red blood cell (RBC), WBC, platelets, and differential WBC counts (187,078 remaining after excluding 7793 individuals). Extreme values of total and differential WBC counts were based on visual inspection and defined as $\log(RBC + 1) \leq 1.4$, $\log(RBC + 1) \geq 2$, $\log(WBC + 1) \leq 1.25$, $\log(WBC + 1) \geq 3$, platelets $\leq 10,000/\mu L$, platelets $\geq 500,000/\mu L$, $\log(\text{neutrophils} + 1) \leq 0.75$, $\log(\text{neutrophils} + 1) \geq 2.75$, $\log(\text{lymphocytes} + 1) \leq 0.10$, $\log(\text{lymphocytes} + 1) \geq 2$, $\log(\text{monocytes} + 1) \geq 0.9$, $\log(\text{eosinophils} + 1) \geq 0.75$, or $\log(\text{basophils} + 1) \geq 0.45$[42]. In addition, we repeated the analysis after excluding 19 individuals who were identified as having mitochondrial disorder based on ICD codes (E88.4, E74.4, G71.3, G93.7, H49.81, H47.22, or G31.82) for hospital admission, and after excluding 312 individuals with any of the 60 identified pathogenic mitochondrial SNVs.

## PheWAS
We used the R package PheWAS[22] to collapse summary diagnoses indicated by primary and secondary ICD-10 disease codes from hospital inpatient records into distinct phecodes comprised of cases and controls for a variety of disease categories. The analysis was restricted to all participants with a MSS including 193,866 individuals and 1618 phecodes with at least one case. This number was used for correction for multiple testing using the Boneferroni method. A logistic regression was run for each phecode regressing the binary phecode onto MSS residuals from a linear regression model adjusting for age, sex, and smoking status. Additional covariates in the logistic regression include age incorporated as a natural spline with four degrees of freedom, sex, and assessment center. Age was modeled using

restricted splines with four degrees of freedom. Firth correction was used to account for unbalanced case and controls when the logistic regression $p$ value was ≤0.0005 using the R package logistf[43].

## PHESANT
We used the PHEnome Scan ANalysis Tool (PHESANT)[23] to identify MSS associated phenotypes in the UKB. Briefly, we tested for the association of mtDNA-CN with ~30,000 traits (Supplementary Data 3), using MSS residuals from a linear regression model adjusting for age, sex, and smoking status, with age incorporated as a natural spline with four degrees of freedom. PHESANT was then run with the residuals with additional adjustment for age, sex, and assessment center.

## Reporting summary
Further information on research design is available in the Nature Portfolio Reporting Summary linked to this article.

## Data availability
Data are available through application to the UK Biobank and Source Data are provided with this paper.

## Code availability
Code for data cleaning and analysis is available on our github repository: https://github.com/ArkingLab. Documentation on MitoHPC pipeline for DNA Nexus server is available in https://github.com/ArkingLab/MitoHPC/blob/main/docs/DNAnexus_CLOUD.md. Documentation on extracting Mitochondrial and NUMT reads from Google Cloud is available in https://github.com/ArkingLab/MitoHPC/blob/main/docs/GOOGLE_CLOUD.md.

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

## Acknowledgements

This research was conducted using the UK Biobank Resource under Application Number 17731. This work was supported by National Heart, Lung and Blood Institute, National Institutes of Health (NIH) grants R01HL144569 (D.E.A) and K24HL148521 (A.A.), National Institute on Aging grants R01AG059727/R01HL15569 (C.L.), NHLBI: R01HL156144 (L.P.G.), National Health and Medical Research Council Fellowship (1159456) and American Australian Association Scholarship (N.J.L.). The content is solely the responsibility of the authors and does not necessarily represent the official views of the NIH. Molecular data for the Trans-Omics in Precision Medicine (TOPMed) program was supported by the National Heart, Lung and Blood Institute (NHLBI). WGS for the Trans-Omics in Precision Medicine (TOPMed) program was supported by the National Heart, Lung and Blood Institute (NHLBI). WGS for "NHLBI TOPMed: Atherosclerosis Risk in Communities (ARIC) (phs001211) was performed at the Baylor College of Medicine Human Genome Sequencing Center (HHSN268201500015C and 3U54HG003273-12S2) and the Broad Institute for MIT and Harvard (3R01HL092577- 06S1). Core support including centralized genomic read mapping and genotype calling, along with variant quality metrics and filtering were provided by the TOPMed Informatics Research Center (3R01HL-117626-02S1; contract HHSN268201800002I). Core support including phenotype harmonization, data management, sample-identity QC, and general program coordination, was provided by the TOPMed Data Coordinating Center (R01HL-120393; U01HL-120393; contract HHSN268201800001I). We gratefully acknowledge the studies and participants who provided biological samples and data for TOPMed. The Genome Sequencing Program (GSP) was funded by the National Human Genome Research Institute (NHGRI), the National Heart, Lung, and Blood Institute (NHLBI), and the National Eye Institute (NEI). The GSP Coordinating Center (U24 HG008956) contributed to cross-program scientific initiatives and provided logistical and general study coordination. The Centers for Common Disease Genomics (CCDG) program was supported by NHGRI and NHLBI, and WGS was performed at the Baylor College of Medicine Human Genome Sequencing Center (UM1 HG008898). The Analysis Commons was funded by R01HL131136. WGS for the Trans-Omics in Precision Medicine (TOPMed) program was supported by the National Heart, Lung and Blood Institute (NHLBI). WGS for "NHLBI TOPMed: Multi-Ethnic Study of Atherosclerosis (MESA)" (phs001416.v1.p1) was performed at the Broad Institute of MIT and Harvard (3U54HG003067-13S1). Centralized read mapping and genotype calling, along with variant quality metrics and filtering were provided by the TOPMed Informatics Research Center (3R01HL-117626-02S1). Phenotype harmonization, data

management, sample-identity QC, and general study coordination, were provided by the TOPMed Data Coordinating Center (3R01HL-120393-02S1), and TOPMed MESA Multi-Omics (HHSN2682015000031/HSN26800004). The MESA projects are conducted and supported by the National Heart, Lung, and Blood Institute (NHLBI) in collaboration with MESA investigators. Support for the Multi-Ethnic Study of Atherosclerosis (MESA) projects are conducted and supported by the National Heart, Lung, and Blood Institute (NHLBI) in collaboration with MESA investigators. Support for MESA is provided by contracts 75N92020D00001, HHSN268201500003I, N01-HC-95159, 75N92020D00005, N01-HC-95160, 75N92020D00002, N01-HC-95161, 75N92020D00003, N01-HC-95162, 75N92020D00006, N01-HC-95163, 75N92020D00004, N01-HC-95164, 75N92020D00007, N01-HC-95165, N01-HC-95166, N01-HC-95167, N01-HC-95168, N01-HC-95169, UL1-TR-000040, UL1-TR-001079, UL1-TR-001420, UL1TR001881, DK063491, and R01HL105756. The authors thank the other investigators, the staff, and the participants of the MESA study for their valuable contributions. A full list of participating MESA investigators and institutes can be found at http://www.mesa-nhlbi.org. WGS for "NHLBI TOPMed: Framingham Heart Study (FHS)" (phs000974) was performed at the Broad Institute of MIT and Harvard (3R01HL092577-06S1 and 3U54HG003067-12S2). Core support including centralized genomic read mapping and genotype calling, along with variant quality metrics and filtering were provided by the TOPMed Informatics Research Center (3R01HL-117626-02S1; contract HHSN268201800002I). Core support including phenotype harmonization, data management, sample-identity QC, and general program coordination, was provided by the TOPMed Data Coordinating Center (R01HL-120393; U01HL-120393; contract HHSN268201800001I). WGS for "NHLBI TOPMed: Women's Health Initiative (WHI)" (phs001237) was performed at the Broad Institute of MIT and Harvard (HHSN268201500014C). Core support including centralized genomic read mapping and genotype calling, along with variant quality metrics and filtering were provided by the TOPMed Informatics Research Center (3R01HL-117626-02S1; contract HHSN268201800002I). Core support including phenotype harmonization, data management, sample-identity QC, and general program coordination was provided by the TOPMed Data Coordinating Center (R01HL-120393; U01HL-120393; contract HHSN268201800001I).

## Author contributions

Y.S.H., S.L.B., W.S, D.P., V.P., J.X., C.L., M.L., and D.E.A. performed the analyses; N.J.L. and M.L. developed the MLC score; N.P. E.G., J.I.R., S.S.R., C.L, D.L., and L.P.G. aided interpretation of the study data; S.L.B., Y.S.H., V.P., M.L.G., and D.E.A. drafted the manuscript; all authors read and approved the final manuscript. J.I.R., S.S.R., C.K., A.R. P.L.A., J.M.M., N.H.C., D.L., and A.A. contributed to study conceptualization and design, and J.I.R., S.S.R., C.K., A.R. P.L.A., J.M.M., N.H.C., M.L.G., D.L., A.A., R.G., and S.D.P aided data collection. E.G. and D.E.A. supervised and managed the study. Y.S.H. and S.L.B. contributed equally to this work.

## Competing interests

The authors declare no competing interests.

## Additional information

[1]McKusick-Nathans Institute, Department of Genetic Medicine, Johns Hopkins University School of Medicine, Baltimore, MD, USA. [2]Department of Natural Sciences, College of Arts and Sciences, Bowie State University, Bowie, MD, USA. [3]Department of Biomedical Engineering, Johns Hopkins University, Baltimore, MD, USA. [4]Department of Laboratory Medicine and Pathology, University of Minnesota, Minneapolis, MN, USA. [5]Department of Genetics, Yale School of Medicine, New Haven, CT, USA. [6]Murdoch Children's Research Institute, Royal Children's Hospital, Melbourne, VIC, Australia. [7]The Institute for Translational Genomics and Population Sciences, Department of Pediatrics, The Lundquist Institute for Biomedical Innovation at Harbor-UCLA Medical Center, Torrance, CA, USA. [8]Center for Public Health Genomics, Department of Public Health Sciences, University of Virginia, Charlottesville, VA, USA. [9]Division of Public Health Sciences, Fred Hutchinson Cancer Research Center, Seattle, WA, USA. [10]Division of Biostatistics, Institute for Health & Equity, and Cancer Center, Medical College of Wisconsin, Milwaukee, WI, USA. [11]Departments of Neurology, Boston University Chobanian & Avedisian School of Medicine, Boston, MA, USA. [12]Framingham Heart Study, Framingham, MA, USA. [13]Department of Biostatistics, School of Public Health, Boston University, Boston, MA, USA. [14]Section of General Internal Medicine, Boston University Chobanian & Avedisian School of Medicine, Boston, MA, USA. [15]National Heart, Lung, and Blood Institute, NIH, Bethesda, MD, USA. [16]Human Genetics Center; Department of Epidemiology, Human Genetics, and Environmental Sciences; School of Public Health, The University of Texas Health Science Center at Houston, Houston, TX, USA. [17]Department of Epidemiology, Rollins School of Public Health, Emory University, Atlanta, GA, USA. [18]Human Genome Sequencing Center, Baylor College of Medicine, Houston, TX, USA. [19]Division of Hematological Malignancies, Sidney Kimmel Comprehensive Cancer Center, Johns Hopkins University, Baltimore, MD, USA. [20]Department of Epidemiology and Medicine, and Welch Center for Prevention, Epidemiology, and Clinical Research, Johns Hopkins University Bloomberg School of Public Health, Baltimore, MD, USA. [21]These authors contributed equally: Yun Soo Hong, Stephanie L. Battle. ✉e-mail: arking@jhmi.edu

