## [Peer Review File · Nature Communications]

REVIEWER COMMENTS

Reviewer #1 (Remarks to the Author): expertise in mitochondrial genetic variants

This manuscript from Hong and colleagues describes an extensive and impressive analysis of whole genome sequencing and matched clinical data from a number of repositories, indicating that mitochondrial heteroplasmy in the blood is associated with all cause mortality, and specifically in worse outcomes for some haematological cancers. This is a substantial and valuable analysis.

Specific comments: the authors might comment on the over-representation of blood-based diseases in their elevated mortality predictions. Does the sample source bias this? Are there data from other tissues which might be used to compare/support these findings?

Related to this, particularly deletions of mtDNA have been associated with diseases of the haematological compartment e.g. Pearson Marrow Pancreas. The authors would significantly strengthen their study by including an analysis of mtDNA deletions in the major cohort. This is not without technical challenges, so I would not say it is a necessary addition - but would make the paper more comprehensive

The link with haematological malignancy is intriguing - particularly as the proportion of cancers with non-synonymous variants in these cancers is diminished compared with solid tumours (see e.g. Yuan et al, 2020, Nature Genetics; Gorelick et al., 2021, Nature Metabolism). Some insight or comment on this would be warranted.

The impact of smoking status on blood heteroplasmy over time is fascinating. Is there any evidence of clonal expansion from the hematopoietic compartment?

For the major effects in the manuscript, particularly all cause mortality in the UK biobank, binning the data by heteroplasmy level would be an instructive analysis. A naive logic would assume that greater heteroplasmies will associate with worse outcomes, however the location of these heteroplasmies and the impact mosaic effects can exert make this an open question.

Reviewer #2 (Remarks to the Author): expertise in bioinformatics and analysis of risk factors

The authors of this study have undoubtedly made significant contributions to the field, and their research findings have the potential to advance our understanding of the association between harmful heteroplasmic mitochondrial mutations and increased risk of overall mortality, particularly cancer-related mortality. Mitochondrial heteroplasmy refers to different variants of mitochondrial DNA within an individual. The accumulation of somatic mutations in mitochondrial DNA contributes to mitochondrial dysfunction, particularly in tissues with high energy demands. Mitochondrial dysfunction has been linked to longevity, cancer, and degenerative diseases.

The study utilized the UK Biobank, a large population-based cohort, to analyze mitochondrial heteroplasmy in 194,871 individuals. To accurately measure mitochondrial DNA single nucleotide variants (SNVs) in large whole-genome sequencing datasets, the researchers developed a bioinformatics pipeline called MitoHPC. MitoHPC constructs a consensus mitochondrial sequence for each individual, enabling more precise read mapping and heteroplasmy measurement. The pipeline accurately identified heteroplasmic SNV variants without generating false-positive or false-negative results in simulated data. Its ability to handle large datasets made it suitable for this study.

The analysis of the UK Biobank data focused on characterizing mtDNA SNVs based on their genetic features and exploring the association between SNVs at highly constrained sites using a novel constraint-based score called Mitochondrial local constraint (MLC) score sum (MSS). The findings were further validated in additional cohorts from the Trans-Omics for Precision Medicine program. Disease phenotypes associated with mtDNA heteroplasmic SNVs were identified, and the detection of these SNVs in blood was proposed as a risk marker for hematological cancers.

The experimental design and methodology employed in this study are robust. The authors have appropriately justified their approach and conducted the experiments with precision. The data analysis is comprehensive and well-presented. The authors have employed appropriate statistical methods to support their conclusions.

Although the manuscript is of high quality and recognizes the importance of bioinformatics pipelines like MitoHPC, I recommend that the authors enhance their description and reasoning of the innovative method used to quantify constraints in mtDNA. The mitochondrial constraint represents the MAIN contribution of the study. Furthermore, a more comprehensive and critical discussion of the results of existing literature and similar methods would significantly improve the manuscript.

Reviewer #3 (Remarks to the Author): expert in biostatistics and risk adjustment

Dear Authors,

The manuscript investigated the association of deleterious heteroplasmic mitochondrial mutations with an increased risk of overall and cancer-specific mortality. I appreciate the authors' efforts in producing valuable research, and I would like to offer some suggestions to further improve its impact and clarity.

There is some ambiguity regarding causal/non-causal relations based on the statements provided by the authors in the paper. In the excerpt, the authors state, "To distinguish between the causal and non-causal models, we incorporated functional annotations to test whether functional heteroplasmies are driving the association between the number of heteroplasmies and risk of mortality." If the authors have compelling evidence for causality, such as temporal precedence of the covariate of main interest along with carefully-controlled variables, it would be beneficial to provide further elaboration on this aspect. However, considering that mutations cannot be regulated but only observed, it might be more appropriate to avoid implying causality and instead emphasize association. With this in mind, I would suggest that the authors review the statements in the Discussion section that mention, "We found... a potential causal role for mitochondrial heteroplasmy in all-cause mortality."

The MLC Score Sum (MSS) is a significant metric for capturing the cumulative mutational burden of mitochondrial heteroplasmy in the manuscript. However, I have some queries regarding the construction of this metric and its scale. It mentions that "The MLC score assigns each heteroplasmy a score ranging from 0-1, with 1 being the most constrained SNV." However, in Figure 5, which illustrates the adjusted dose-response relationship between MSS and all-cause mortality, the range of MSS appears to be from 0 to 2, with a caption stating that "58 participants with MSS greater than 2 were excluded from the plot" implying that the range of MSS may be beyond 2. To gain a better understanding, it would be helpful if the authors could provide further clarification on the MSS.

For some models, age or MSS were included as 'restricted cubic splines with 4 degrees of freedom'. However, there is limited explanation provided as to why non-linear terms were necessary. Furthermore, it is observed that MSS was adjusted as-is for some other models, allowing for the interpretation of a 1-unit increase in regression coefficients (evident, for instance, in Figure 4). This selective or inconsistent use of covariates warrants clarification, accompanied by sufficient reasoning to justify such approaches. Additionally, it appears that the authors used time-on-study (follow-up time since study entry or disease diagnosis) as the time scale when employing survival models, Cox or subdistribution hazards models. However, considering that age might strongly influence the outcome of interest (mortality or cancer disease), I would recommend exploring the use of age as a time scale in the models. This adjustment could provide valuable insights and enhance the understanding of the relationship between age and the outcome under investigation, rather than estimating the non-linear impact of age as a model covariate.

There is a query regarding the meaning of 'the delta method' mentioned in the statements, "We further estimated the prevalence ratios (PRs) of the 4 types of cancers using logistic regression models. ... The corresponding 95% CIs were estimated using the delta method." If the delta method refers to an approximation of the standard error (SE) computed from a standard tool for logistic regression models, it may not require special mention in this context. However, if it entails a different approach, further clarification is necessary to understand how the confidence intervals were obtained.

The inclusion of details on how the data were restricted, as indicated in the statement "...further restricted the study population to participants without extreme values of red blood cell (RBC), WBC, and differential WBC counts. Extreme values of total and differential WBC counts were based on inspection and defined as ..." is commendable. However, it would be beneficial to provide information on the number of individuals excluded from the study population as a result of these restrictions. This will help assess the potential impact of the exclusions and evaluate the generalizability of the findings. Furthermore, it is important to clarify whether the determination of extreme values was based on statistical analyses or common domain knowledge. Providing insights into the rationale behind selecting those specific cut-off values will enhance the transparency and reproducibility of the study. If the cut-off values were determined through statistical analysis, such as outlier detection methods or reference ranges, it would be valuable to mention the approach employed. Alternatively, if they were derived from established clinical guidelines or previous research, this should also be clarified.

In conclusion, the manuscript demonstrates several commendable aspects and valuable contributions to the field. The authors should be commended for their meticulous research efforts and the insights they have provided. Addressing further clarification and explanation as commented above will enhance the overall impact and clarity of the manuscript.

We greatly appreciate the reviewer's comments and have substantially updated the manuscript as detailed below.

Reviewer #1 (Remarks to the Author): expertise in mitochondrial genetic variants

This manuscript from Hong and colleagues describes an extensive and impressive analysis of whole genome sequencing and matched clinical data from a number of repositories, indicating that mitochondrial heteroplasmy in the blood is associated with all cause mortality, and specifically in worse outcomes for some haematological cancers. This is a substantial and valuable analysis.

Thank you for your comments.

Specific comments:

1. The authors might comment on the over-representation of blood-based diseases in their elevated mortality predictions. Does the sample source bias this? Are there data from other tissues which might be used to compare/support these findings?

As the reviewer suggests, it is likely that the DNA source results in an over-representation of blood-based diseases, particularly hematological cancers, for which blood cells are the relevant tissue. Despite this over-representation, we still see significant associations with phenotypes not directly related to blood (e.g. lung and breast cancer, chronic bronchitis). As we only had WGS data derived from blood, we were not able to determine the association of MSS measured in other tissues with mortality or cancer risk. However, various studies that have looked directly at cancer tissues have identified the presence of excess functional mtDNA heteroplasmies in various cancers, including lung and breast cancers^{1,2}, supporting the idea that mtDNA heteroplasmy detected in blood may be a viable biomarker of additional cancer types. Finally, we note the high-relevance of the availability of a readily accessible tissue like blood for assessing disease risk. In response, we have made the following changes to the Discussion section of the manuscript:

“While we were limited to mtDNA heteroplasmy measured in blood and, thus, were not able to evaluate the association of MSS measured in additional tissues with mortality or cancer risk, various studies have found an excess of functional mtDNA heteroplasmies in many cancers, including lung and breast cancers tissues^{1,2}.

A recent comprehensive study from the International Cancer Genome/The Cancer Genome Atlas Pan-Cancer Analysis of Whole Genome (PCAWG) Consortium² looking at WGS from 2,658 cancers across 38 tumor types provides compelling evidence for enrichment of mtDNA truncating mutations in several cancer types, including kidney, colorectal, and thyroid cancers. Similarly, a study of 10,132 paired tumor/normal samples with whole exome sequencing from The Cancer Genome Atlas (TCGA) identified 4,381 mtDNA mutations and various levels of truncating mutations in several cancer types, including both hematologic and solid cancers¹. These studies support the idea that mtDNA heteroplasmy detected in blood may be a viable biomarker for even some non-hematologic cancers. However, there have been a limited number of studies linking mitochondrial heteroplasmy in blood and hematologic malignancies. Intriguingly, while of marginal significance, one study demonstrated that patients with clinically refractive leukemia tended to have higher rates of amino acid altering mutations.

Accordingly, regular screening for cancer-causing mtDNA mutations or SNVs at highly constrained sequences (high MLC score) could be developed as a prognostic marker for cancer patients and surveillance tool for cancer survivors.”

2. Related to this, particularly deletions of mtDNA have been associated with diseases of the haematological compartment e.g. Pearson Marrow Pancreas. The authors would significantly strengthen their study by including an analysis of mtDNA deletions in the major cohort. This is not without technical challenges, so I would not say it is a necessary addition - but would make the paper more comprehensive.

In the UK Biobank samples, we recently identified 2 large-fragment deletions (314_955del and 8482_13446del). Of the 194,871 individuals evaluated, 314_955del was present in 72.5% (n = 141,420) and 8482_13446del in 0.05% (n = 89). In our preliminary analysis, we did not find any association between the deletions and hematologic cancers, and we have not included the results as we believed they are still somewhat preliminary, and do not meaningfully impact the results presented in the current manuscript. We would be happy to include the findings if the editor and the reviewer deem it necessary.

3. The link with haematological malignancy is intriguing - particularly as the proportion of cancers with non-synonymous variants in these cancers is diminished compared with solid tumours (see e.g. Yuan et al, 2020, Nature Genetics; Gorelick et al., 2021, Nature Metabolism). Some insight or comment on this would be warranted.

The main difference between our manuscript and those cited by the reviewer is that those studies have looked directly at cancerous tissue, while we have assessed heteroplasmy in blood, and this is presumably a non-cancerous tissue even for the hematological cancers (cell counts did not indicate active cancer in UKB participants). We also focused on incident events, demonstrating the utility of heteroplasmy as a prognostic marker for mortality/disease risk. Unfortunately, we do not have any great insight into what drives the burden of mitochondrial mutations in cancerous tissue across cancer types, which was not a focus of our manuscript, and can only speculate that hematological cancers may have a lower threshold for mitochondrial dysfunction contributing to development of cancer, or alternatively, that too high a burden of mutation may limit the progression of hematological cancers in same way. This would certainly be an important question for future studies.

4. The impact of smoking status on blood heteroplasmy over time is fascinating. Is there any evidence of clonal expansion from the hematopoietic compartment?

Our working hypothesis is that given the VAF cut-off of 5%, all heteroplasmies that we detect are an indication of clonal expansion, and that this clonal expansion likely occurs in the hematopoietic compartment. We have no direct evidence for this but note that heteroplasmic VAF appears to be remarkably stable over time based on resequencing ~800 population-based samples at 2 timepoints 10 years apart (R=0.989) (manuscript in preparation), indicating that the heteroplasmy is found in HSCs. In further support of this hypothesis, we note that previously published single cell analyses found heteroplasmy across blood cell types³.

5. For the major effects in the manuscript, particularly all cause mortality in the UK biobank, binning the data by heteroplasmy level would be an instructive analysis. A naive logic would assume that greater heteroplasmy will associate with worse outcomes, however the location of these heteroplasmy and the impact mosaic effects can exert make this an open question.

As the reviewer noted, the evaluation of the effect of heteroplasmy levels (i.e., different variant allele fractions [VAF]) on outcomes is complex as each variant will exhibit different heteroplasmy levels, complicating the assignment of a single heteroplasmy level to each individual. In response, we have evaluated the association of VAF on all-cause mortality under three different scenarios: 1) VAF of a randomly selected heteroplasmy SNV in each individual; 2) VAF of the heteroplasmy SNV with the largest MLC score in each individual; and 3) VAF of variants in individuals who carry a single variant (Singleton). In all three scenarios, an increase in VAF was not associated with higher mortality, whereas MLC score sum (MSS) was consistently associated with a higher risk of mortality (**Table R1**). These findings suggest that MSS is more relevant for disease risk than heteroplasmy VAF. Additionally, we have observed that more deleterious mutations tend to have lower VAF (Fig. 2F), potentially because they are less tolerated, providing evidence that MSS needs to be accounted for when evaluating the impact of VAF on outcomes. Of note, also, we have defined heteroplasmy at a threshold of 5% and, thus, may have missed any effects that occur at lower VAF.

Table R1. Hazard ratios (95% confidence intervals) for all-cause mortality by variant allele fraction (VAF) and MLC score sum (MSS).

	VAF only	MSS only	Both VAF and MSS	
	VAF (per 10%)	MSS (per 1-unit)	VAF	MSS
Randomly selected heteroplasmy SNV	0.99 (0.98, 1.01)	1.32 (1.21, 1.44)	0.99 (0.98, 1.01)	1.30 (1.20, 1.41)
Heteroplasmy SNV with the largest MLC score	0.99 (0.98, 1.00)	1.32 (1.21, 1.44)	1.00 (0.99, 1.01)	1.31 (1.20, 1.42)
Singleton	0.99 (0.98, 1.00)	1.40 (1.22, 1.61)	0.99 (0.98, 1.01)	1.33 (1.17, 1.51)

*All models are adjusted for age, sex, smoking status, and stratified by center.

In response, we have added the results as **Extended Data Table 4** and revised the Main Text, Discussion, and Methods as follows:

In the Main Text:

“In addition, we evaluated whether VAF is associated with all-cause mortality and did not observe any association between VAF and mortality (**Extended Data Table 4**), suggesting that MSS is more relevant for disease risk than VAF.”

In the Discussion:

“We also found MSS to be a better predictor of mortality than heteroplasmy level, or VAF, which is in line with our observation that more deleterious mutations tend to have lower VAF, potentially because they are less tolerated. We, however, may have missed any effects that occur at lower VAF, as we have defined heteroplasmy at a threshold of 5%.”

In the Methods:

“Models for evaluating the association between heteroplasmy count and all-cause mortality were stratified by assessment center and were adjusted for age using restricted cubic splines with 4 degrees of freedom, sex, and smoking status (never, former, or current). Models evaluating the association between VAF and all-cause mortality included the same set of covariates. However, because each variant exhibits different heteroplasmy VAF, complicating the assignment of a single VAF for each individual, we evaluated the association of VAF on all-cause mortality under three different scenarios: 1) VAF of a randomly selected heteroplasmic SNV in each individual; 2) VAF of the heteroplasmic SNV with the largest MLC score in each individual; and 3) VAF of variants in individuals who carry a single variant. We further adjusted for MSS and compared any change in the estimates before and after adjustment.”

Reviewer #2 (Remarks to the Author): expertise in bioinformatics and analysis of risk factors

The authors of this study have undoubtedly made significant contributions to the field, and their research findings have the potential to advance our understanding of the association between harmful heteroplasmic mitochondrial mutations and increased risk of overall mortality, particularly cancer-related mortality. Mitochondrial heteroplasmy refers to different variants of mitochondrial DNA within an individual. The accumulation of somatic mutations in mitochondrial DNA contributes to mitochondrial dysfunction, particularly in tissues with high energy demands. Mitochondrial dysfunction has been linked to longevity, cancer, and degenerative diseases.

The study utilized the UK Biobank, a large population-based cohort, to analyze mitochondrial heteroplasmy in 194,871 individuals. To accurately measure mitochondrial DNA single nucleotide variants (SNVs) in large whole-genome sequencing datasets, the researchers developed a bioinformatics pipeline called MitoHPC. MitoHPC constructs a consensus mitochondrial sequence for each individual, enabling more precise read mapping and heteroplasmy measurement. The pipeline accurately identified heteroplasmic SNV variants without generating false-positive or false-negative results in simulated data. Its ability to handle large datasets made it suitable for this study.

The analysis of the UK Biobank data focused on characterizing mtDNA SNVs based on their genetic features and exploring the association between SNVs at highly constrained sites using a novel constraint-based score called Mitochondrial local constraint (MLC) score sum (MSS). The findings were further validated in additional cohorts from the Trans-Omics for Precision Medicine program. Disease phenotypes associated with mtDNA heteroplasmic SNVs were identified, and the detection of these SNVs in blood was proposed as a risk marker for hematological cancers.

The experimental design and methodology employed in this study are robust. The authors have appropriately justified their approach and conducted the experiments with precision. The data analysis is comprehensive and well-presented. The authors have employed appropriate statistical methods to support their conclusions.

Thank you for your comments.

1. Although the manuscript is of high quality and recognizes the importance of bioinformatics pipelines like MitoHPC, I recommend that the authors enhance their description and reasoning of the innovative method used to quantify constraints in mtDNA. The mitochondrial constraint represents the MAIN contribution of the study. Furthermore, a more comprehensive and critical discussion of the results of existing literature and similar methods would significantly improve the manuscript.

Mitochondrial local constraint (MLC) score is a measure of local tolerance to base or amino acid substitutions and is described in detail in a separate manuscript under review at *Nature* and posted on bioRxiv (<https://doi.org/10.1101/2022.12.16.520778>)⁴. Briefly, an observed:expected ratio of substitutions is calculated for every possible mtDNA SNV by applying a sliding window method and percentile-ranked to a score ranging from 0 to 1, with 0 being the least constrained and 1 being the most constrained. We chose MLC score for our analysis as it overcomes the limitations in the algorithm currently recommended by the ACMG/AMP guidelines⁵ for mtDNA

variants, APOGEE⁶, by incorporating variants in non-coding regions, tRNA, and rRNA, as well as non-missense mutations. In addition, a constraint model quantifies the removal of deleterious variation from the population by selection, thereby reflecting the functional importance of each variant across the entire mtDNA genome. In response, we have introduced the following changes to the Main Text:

“Second, we incorporated functional annotation into our models to test whether functional heteroplasmies drive the association between the number of heteroplasmies and risk of mortality. We used the mitochondrial local constraint (MLC) score⁴, a recently developed annotation metric based on local constraint, which quantifies the local tolerance to base or amino acid substitutions for each base pair across the entire mtDNA genome. The MLC score assigns each heteroplasmy a score ranging from 0 to 1, with 1 being the most constrained SNV, and therefore, likely to have the most deleterious impact when mutated. The MLC score addresses the limitations in the algorithm currently recommended by the ACMG/AMP guidelines⁵ for mtDNA variants by incorporating variants in non-coding regions, tRNA, and rRNA, as well as non-missense mutations. We randomly selected one heteroplasmic SNV from each participant, and then tested whether the MLC score had a significant association with all-cause mortality. MLC score was significantly associated with mortality (adjusted HR comparing MLC score of 1 to 0 was 1.34; 95% CI 1.21, 1.49; $P = 5.4 \times 10^{-8}$; **Extended Data Fig. 4b**), while continuing to adjust for center (stratification factor), age, sex, smoking status, and the number of heteroplasmies.”

Furthermore, we performed additional analysis for the association between APOGEE score and all-cause mortality for a heteroplasmic SNV randomly selected in each individual. The hazard ratio (HRs) and corresponding 95% confidence interval (CI) comparing an APOGEE score of 1 to 0 was 1.20 (95% CI 1.06, 1.37) after adjusting for age (restricted cubic splines with 4 degrees of freedom), sex, and smoking status, and stratified by center (**Figure R1**). However, after additionally adjusting for MLC score of the selected heteroplasmic SNV, APOGEE was no longer associated with mortality (adjusted HR 0.96; 95% CI 0.82, 1.12). Similarly, while APOGEE summed across all missense mutations was associated with a higher risk of mortality (aHR 1.17; 95% CI 1.04, 1.31), this association disappeared after additionally accounting for MLC score sum (MSS). Collectively, the results suggest that the MLC score is a better functional marker for mtDNA variants than APOGEE score. However, as the two metrics are not directly comparable approaches, with MLC trained specifically on heteroplasmic variants, we have not included these results in the manuscript. We would be happy to consider the inclusion of these findings upon the editor’s or the reviewer’s request.

Figure R1. Hazard ratios (95% confidence intervals) for APOGEE and all-cause mortality. For single variant analyses, we randomly selected one heteroplasmic SNV in each individual and evaluated the association of APOGEE score and MLC score for that variant separately with all-cause mortality using Cox proportional hazards models stratified by center and adjusted for age, sex, and smoking status. We then included both APOGEE score and MLC score for the given variant in the same model (“Both”). For all variants analyses, we used APOGEE summed across all missense mutations (APOGEE sum) and MLC score sum (MSS) calculated for each individual. We first evaluated the associations of APOGEE sum and MSS with all-cause mortality separately and then jointly in the same model. Hazard ratios for all-cause mortality were estimated using Cox proportional hazards models stratified by center and adjusted for age, sex, and smoking status.

Reviewer #3 (Remarks to the Author): expert in biostatistics and risk adjustment

Dear Authors,

The manuscript investigated the association of deleterious heteroplasmic mitochondrial mutations with an increased risk of overall and cancer-specific mortality. I appreciate the authors' efforts in producing valuable research, and I would like to offer some suggestions to further improve its impact and clarity.

Thank you for your comments.

1. There is some ambiguity regarding causal/non-causal relations based on the statements provided by the authors in the paper. In the excerpt, the authors state, "To distinguish between the causal and non-causal models, we incorporated functional annotations to test whether functional heteroplasmies are driving the association between the number of heteroplasmies and risk of mortality." If the authors have compelling evidence for causality, such as temporal precedence of the covariate of main interest along with carefully-controlled variables, it would be beneficial to provide further elaboration on this aspect. However, considering that mutations cannot be regulated but only observed, it might be more appropriate to avoid implying causality and instead emphasize association. With this in mind, I would suggest that the authors review the statements in the Discussion section that mention, "We found... a potential causal role for mitochondrial heteroplasmy in all-cause mortality."

Thank you for this comment. We have revised the statement as follows:

"We found that the presence of heteroplasmy was associated with all-cause mortality and that a function score based on local sequence constraint, MSS, was a better predictor than the number of heteroplasmies, implicating a potential role for mitochondrial heteroplasmy in all-cause mortality."

2. The MLC Score Sum (MSS) is a significant metric for capturing the cumulative mutational burden of mitochondrial heteroplasmy in the manuscript. However, I have some queries regarding the construction of this metric and its scale. It mentions that "The MLC score assigns each heteroplasmy a score ranging from 0-1, with 1 being the most constrained SNV." However, in Figure 5, which illustrates the adjusted dose-response relationship between MSS and all-cause mortality, the range of MSS appears to be from 0 to 2, with a caption stating that "58 participants with MSS greater than 2 were excluded from the plot" Implying that the range of MSS may be beyond 2. To gain a better understanding, it would be helpful if the authors could provide further clarification on the MSS.

The MLC score is assigned for each heteroplasmic variant and ranges from 0 to 1, with 1 being the most constrained SNV. The MLC Score Sum (MSS), on the other hand, is generated to incorporate the total burden of heteroplasmies for each individual and is the sum of all MLC scores within a given individual. Therefore, MSS may be greater than 1. We revised the corresponding description in the Main Text as follows:

“To capture the impact of multiple heteroplasms, we generated an MLC score sum (MSS) by summing all MLC scores within a given individual.”

3. For some models, age or MSS were included as 'restricted cubic splines with 4 degrees of freedom'. However, there is limited explanation provided as to why non-linear terms were necessary. Furthermore, it is observed that MSS was adjusted as-is for some other models, allowing for the interpretation of a 1-unit increase in regression coefficients (evident, for instance, in Figure 4). This selective or inconsistent use of covariates warrants clarification, accompanied by sufficient reasoning to justify such approaches.

Thank you for the opportunity to clarify the models we used for age and MSS. In our descriptive analysis, age was non-linearly associated with MSS (**Figure R2**) and, therefore, we used restricted cubic splines with 4 degrees of freedom to model the non-linear association flexibly consistently across all models. For MSS, we used restricted cubic splines with 4 degrees of freedom only for evaluating a dose-response relationship in Figure 5. The p-value for non-linearity was 0.55 and, thus, for all other analyses, MSS was modeled linearly and can be interpreted as 1-unit increase in MSS. In response, we revised the corresponding sentence as follows:

“After additionally adjusting for alcohol intake, body mass index (BMI), white blood cell (WBC) counts, and haplogroup, MSS was associated with all-cause mortality in a dose-response manner (**Fig. 5**; P for non-linearity = 0.55) and a 1-unit increase in MSS was associated with a 28% (aHR 1.28; 95% CI 1.20, 1.37) increase in the risk of mortality.”

Figure R2. The association between age and MLC score sum.

4. Additionally, it appears that the authors used time-on-study (follow-up time since study entry or disease diagnosis) as the time scale when employing survival models, Cox or subdistribution hazards models. However, considering that age might strongly influence the outcome of interest (mortality or cancer disease), I would recommend exploring the use of age as a time scale in the models. This adjustment could provide valuable insights and enhance the understanding of the relationship between age and the outcome under investigation, rather than estimating the non-linear impact of age as a model covariate.

In response, we explored the use of age as a time scale in the association of MSS and all-cause mortality. The HR was 1.27 (95% CI 1.19, 1.36) after adjusting for sex and smoking status, and stratified by center, and was almost identical to the results using a non-linear term for age (aHR

1.28 (95% CI 1.20, 1.37)). We believe that this finding suggests that including a restricted cubic spline term for age, as previously done, appropriately accounts for the non-linear impact of age as a model covariate. We did not introduce any change to the manuscript.

5. There is a query regarding the meaning of 'the delta method' mentioned in the statements, "We further estimated the prevalence ratios (PRs) of the 4 types of cancers using logistic regression models. ... The corresponding 95% CIs were estimated using the delta method." If the delta method refers to an approximation of the standard error (SE) computed from a standard tool for logistic regression models, it may not require special mention in this context. However, if it entails a different approach, further clarification is necessary to understand how the confidence intervals were obtained.

As the reviewer noted, the delta method refers to an approximation of the standard error (SE) computed from logistic regression models. After running logistic regression models, we calculated prevalence ratios (PR) using the marginally predicted probabilities associated with a 1-unit increase in MSS and the 95% confidence intervals for PRs were estimated using the delta method. In response, we removed the mention of delta method.

6. The inclusion of details on how the data were restricted, as indicated in the statement "...further restricted the study population to participants without extreme values of red blood cell (RBC), WBC, and differential WBC counts. Extreme values of total and differential WBC counts were based on inspection and defined as ..." is commendable. However, it would be beneficial to provide information on the number of individuals excluded from the study population as a result of these restrictions. This will help assess the potential impact of the exclusions and evaluate the generalizability of the findings. Furthermore, it is important to clarify whether the determination of extreme values was based on statistical analyses or common domain knowledge. Providing insights into the rationale behind selecting those specific cut-off values will enhance the transparency and reproducibility of the study. If the cut-off values were determined through statistical analysis, such as outlier detection methods or reference ranges, it would be valuable to mention the approach employed. Alternatively, if they were derived from established clinical guidelines or previous research, this should also be clarified.

There were 7793 individuals excluded from the analysis due to extreme values of blood cell counts. The exclusion criteria was based on common domain knowledge and, in response, we have introduced the changes as follows:

“Moreover, we further restricted the study population to participants without extreme values of red blood cell (RBC), WBC, platelets, and differential WBC counts (187,078 remaining after excluding 7793 individuals). Extreme values of total and differential WBC counts were based on visual inspection and defined as $\log(\text{RBC} + 1) \leq 1.4$, $\log(\text{RBC} + 1) \geq 2$, $\log(\text{WBC} + 1) \leq 1.25$, $\log(\text{WBC} + 1) \geq 3$, $\text{platelets} \leq 10,000/\mu\text{L}$, $\text{platelets} \geq 500,000/\mu\text{L}$, $\log(\text{neutrophils} + 1) \leq 0.75$, $\log(\text{neutrophils} + 1) \geq 2.75$, $\log(\text{lymphocytes} + 1) \leq 0.10$, $\log(\text{lymphocytes} + 1) \geq 2$, $\log(\text{monocytes} + 1) \geq 0.9$, $\log(\text{eosinophils} + 1) \geq 0.75$, or $\log(\text{basophils} + 1) \geq 0.45$ ⁷.

7. In conclusion, the manuscript demonstrates several commendable aspects and valuable contributions to the field. The authors should be commended for their meticulous research efforts and the insights they have provided. Addressing further clarification and explanation as commented above will enhance the overall impact and clarity of the manuscript.

Thank you again for your comments.

References

1. Gorelick, A. N. *et al.* Respiratory complex and tissue lineage drive recurrent mutations in tumour mtDNA. *Nat Metab* **3**, 558–570 (2021).
2. Yuan, Y. *et al.* Comprehensive molecular characterization of mitochondrial genomes in human cancers. *Nat Genet* **52**, 342–352 (2020).
3. Walker, M. A. *et al.* Purifying Selection against Pathogenic Mitochondrial DNA in Human T Cells. *N Engl J Med* **383**, 1556–1563 (2020).
4. Lake, N. J. *et al.* Quantifying constraint in the human mitochondrial genome. 2022.12.16.520778 Preprint at <https://doi.org/10.1101/2022.12.16.520778> (2023).
5. McCormick, E. M. *et al.* Specifications of the ACMG/AMP standards and guidelines for mitochondrial DNA variant interpretation. *Hum Mutat* **41**, 2028–2057 (2020).
6. Castellana, S. *et al.* High-confidence assessment of functional impact of human mitochondrial non-synonymous genome variations by APOGEE. *PLoS Comput Biol* **13**, e1005628 (2017).
7. Yang, S. Y. *et al.* Blood-derived mitochondrial DNA copy number is associated with gene expression across multiple tissues and is predictive for incident neurodegenerative disease. *Genome Res* **31**, 349–358 (2021).

REVIEWERS' COMMENTS

Reviewer #1 (Remarks to the Author):

The authors have addressed my comments.

I would be minded to include the data previously held back on deletions - that such a vast number of samples contain the mtDNA common deletion, presumably at very low VAF, is quite staggering to me, and would be of interest to the mtDNA community.

If this could be reported, alongside the VAFs, it would be a valuable addition.

Reviewer #2 (Remarks to the Author):

Dear Editor,

They have expanded the methodological section, providing some reasoning for the procedures used in the study. They have provided further insights into data analysis methods, acknowledging the importance of robust statistical approaches.

I believe that these revisions have significantly strengthened the manuscript, making it more accessible and informative to the readership of Nature Communications.

Sincerely,

Reviewer #3 (Remarks to the Author):

I appreciate that the authors clarified my questions and comments. I do not have further comments on the manuscript. Sincerely,

Reviewer #1 (Remarks to the Author):

The authors have addressed my comments.

I would be minded to include the data previously held back on deletions - that such a vast number of samples contain the mtDNA common deletion, presumably at very low VAF, is quite staggering to me, and would be of interest to the mtDNA community.

If this could be reported, alongside the VAFs, it would be a valuable addition.

Thank you. In response, we have added the findings on mtDNA deletions as follows:

In the Main Text:

“We have additionally identified 2 large-fragment deletions (314-955del and 8482-13446del) in the UKB participants. Of the 194,871 individuals evaluated, 314-955del was present in 72.5% (n = 141,420) and 8482-13446del in 0.05% (n = 89). The VAF for both deletions were low, with the median (range) of 0.24% (0.08–3.88%) for 314-955del and 0.12% (0.10–0.28%) for 8482-13446del.”

In the Methods:

“Deletions in the mtDNA were identified by analyzing split read alignments. Split reads are reads from a unique region of the read that have two or more alignments to the reference with the same orientation. We identified the alignment end positions on chrM (1st alignment 3' & 2nd alignment 5') and counted their frequencies. A minimum frequency of 2 was used. We also confirmed that none of the split reads had a mate mapped to other chromosomes, which could be a NUMT indicator.”

Reviewer #2 (Remarks to the Author):

Dear Editor,

They have expanded the methodological section, providing some reasoning for the procedures used in the study. They have provided further insights into data analysis methods, acknowledging the importance of robust statistical approaches.

I believe that these revisions have significantly strengthened the manuscript, making it more accessible and informative to the readership of Nature Communications.

Thank you.

Reviewer #3 (Remarks to the Author):

I appreciate that the authors clarified my questions and comments. I do not have further comments on the manuscript.

Thank you.